# Identification of a neurocircuit underlying regulation of feeding by stress-related emotional responses

Yuanzhong Xu[1,7], Yungang Lu [1,7], Ryan M. Cassidy [1,2,7], Leandra R. Mangieri[1,2], Canjun Zhu[1], Xugen Huang[1], Zhiying Jiang[1], Nicholas J. Justice[1,2], Yong Xu [3], Benjamin R. Arenkiel[4,5] & Qingchun Tong [1,2,6]

Feeding is known to be profoundly affected by stress-related emotional states and eating disorders are comorbid with psychiatric symptoms and altered emotional responses. The neural basis underlying feeding regulation by stress-related emotional changes is poorly understood. Here, we identify a novel projection from the paraventricular hypothalamus (PVH) to the ventral lateral septum (LSv) that shows a scalable regulation on feeding and behavioral changes related to emotion. Weak photostimulation of glutamatergic PVH→LSv terminals elicits stress-related self-grooming and strong photostimulation causes fear-related escape jumping associated with respective weak and strong inhibition on feeding. In contrast, inhibition of glutamatergic inputs to LSv increases feeding with signs of reduced anxiety. LSv-projecting neurons are concentrated in rostral PVH. LSv and LSv-projecting PVH neurons are activated by stressors in vivo, whereas feeding bouts were associated with reduced activity of these neurons. Thus, PVH→LSv neurotransmission underlies dynamic feeding by orchestrating emotional states, providing a novel neural circuit substrate underlying comorbidity between eating abnormalities and psychiatric disorders.

[1] Brown Foundation Institute of Molecular Medicine, University of Texas McGovern Medical School, Houston, TX 77030, USA. [2] Graduate Program in Neuroscience of the University of Texas MD Anderson UTHealth Graduate School of Biomedical Sciences, Houston, TX 77030, USA. [3] Children's Nutrition Research Center, Department of Pediatrics, Baylor College of Medicine, One Baylor Plaza, Houston, TX 77030, USA. [4] Department of Molecular & Human Genetics, Baylor College of Medicine, Houston, TX 77030, USA. [5] Department of Neuroscience and Jan Duncan Neurological Research Institute, Texas Children's Hospital, Houston, TX 77030, USA. [6] Department of Neurobiology and Anatomy, University of Texas McGovern Medical School, Houston, TX 77030, USA. [7] These authors contributed equally: Yuanzhong Xu, Yungang Lu, Ryan M. Cassidy. Correspondence and requests for materials should be addressed to Q.T. (email: qingchun.tong@uth.tmc.edu)

Feeding is a complex behavior and known to be profoundly affected by stress-related emotional states[1]. Research in animals suggests that motivation of feeding competes with emotional states encoding for potential environmental dangers[2]. As such, acute changes in feeding behaviors are associated with adaptive changes in emotional states related to stress and anxiety[3,4]. Despite differences in describing behaviors related to emotional states between rodents and humans, various states in valence related to emotion are conversed between rodents and humans[5]. In humans, chronic eating disorders are accompanied by marked maladaptive changes in emotional state and are known to be associated with altered stress and anxiety[6–9], whereas food over-consumption is often associated with increased impulsivity and positive reinforcement, similar to drug addiction[10,11]. These observations support the idea that regulation of feeding and emotional states involves a common brain neural pathway. However, the neural basis underlying behavioral connections between feeding and emotional states is poorly understood.

Emerging evidence suggests that the hypothalamic regulation of feeding involves changes in valence related to emotional states. Thus, feeding elicited by activation of agouti-related protein (AgRP) neurons is associated with changes in aversion and signs for anxiety[12–16]. In addition, activation of lateral hypothalamus (LH) GABAergic neurons promotes feeding associated with positive reinforcement, whereas activation of LH glutamatergic neurons inhibits feeding and promotes negative valence[17–19]. Conversely, recent results have identified that the amygdala, a main brain region in emotion control, plays an important role in feeding regulation[20,21]. These observations, taken together, support a duel role in the regulation of feeding and behaviors related to emotion changes for both brain feeding and emotion centers.

The paraventricular hypothalamus (PVH), an important brain site for homeostatic regulation, coordinates numerous behavioral, and physiological adaptions for survival, including feeding[22]. Extensive studies on PVH projection patterns and circuit function suggest that PVH neurons regulate feeding through various projections to within the hypothalamus or downstream brain stem and hindbrain neurons[1,23–29]. Among various subtypes of PVH neurons, those expressing corticotrophin-releasing hormone (CRH neurons) are known to initiate the neuroendocrine aspect of stress responses; however, this response is not involved in stress-induced hypophagia[30,31]. It is not clear how PVH neurons integrate stress-related responses to feeding control. Importantly, despite previous studies implicating connections between hypothalamic periventricular neurons and basal forebrain structures[32], where lie brain centers for emotion control, a functional demonstration on PVH projections to these brain structures is still lacking.

Our previous study showed that PVH neuron activity controlled by GABAergic and glutamatergic projections from LH determines feeding versus self-grooming, a typical stress-related compulsive behavior[33]. Here, we identify a novel projection from the PVH to the ventral lateral septum (LSv), a basal forebrain region known to regulate fear and aggression[32,34]. We found that photostimulation of glutamatergic PVH→LSv terminals produced a scalable effect on feeding inhibition and behavioral signs of stress and fear, highlighted by respective self-grooming and frantic escape jumping, whereas inhibition of LSv glutamate receptors increased feeding behavior with signs of reduced anxiety. LSv and LSv-projecting PVH neurons were activated by environmental stressors in vivo, whereas feeding was prompted with reduced activity of these neurons. Our results also demonstrated that mice preferred to hunger by avoiding stimulation of the PVH→LSv projection, suggesting that emotional states orchestrated by the stimulation inhibit hunger, reminiscent of restrictive feeding in anorexia nervosa. Thus, we identify a novel neurocircuit in which PVH neurons regulate feeding by orchestrating emotional states by integrating environmental stressors by direct glutamatergic projections to the LSv, thus providing a candidate neural circuit substrate for comorbidity between eating and psychiatric disorders.

## Results

**PVH provides monosynaptic glutamatergic projections to LSv.** To map novel downstream sites of PVH neurons, we genetically target PVH neurons using previously described Sim1-Cre mice[35]. Four weeks after delivery of Cre-dependent adeno-associated viral (AAV) vectors engineered to express a channelrhodopsin (ChR2)-eGFP fusion protein (AAV-Flex-ChR2-eGFP) to the PVH of *Sim1-Cre::ROSA-lsl-tdTomato* mice (Fig. 1a, b), we assayed for ChR2 projections originating from Sim1 expressing PVH neurons in potential downstream target domains. Notably, we identified discrete GFP-expressing fibers in the ventral part of lateral septum (LSv) (Fig. 1c), a known brain region for fear and aggression[34]. The LSv region with GFP fibers was found to largely span from Bregma + 1.18 to +0.98 (Supplementary Fig. S1a–f). It is interesting to note that the LSv was the only region at these Bregma levels found to receive abundant PVH inputs (Supplementary Fig. S1g–j). To determine whether these projections were functional, we next recorded LSv neurons for excitatory postsynaptic currents (EPSCs) following photostimulation of ChR2-expressing hypothalamic fibers. 12 out of 18 recorded LSv neurons showed strong photo-induced postsynaptic responses, all of which were blocked by the glutamate receptor antagonists (CNQX/APV) (Fig. 1d), and refractory to the GABA-A receptor antagonist (GABAzine) (Fig. 1e) and AP4/TTX (Fig. 1f), suggesting that the PVH provides monosynaptic glutamatergic projections to LSv neurons. In the same preparation, we also recorded inhibitory IPSCs (12 out of 15), all of which were blocked by the GABA-A receptor blocker GABAzine (Fig. 1g). Surprisingly, IPSCs in most neurons were also blocked by CNQX/APV (10 out of 12) (Fig. 1h) and 4-AP/TTX (8 out of 12) (Fig. 1i). Moreover, the latency for IPSCs was significantly longer than that of EPSCs (Supplementary Fig. S1k). These data imply that the majority of recorded IPSCs were generated by activation of local GABAergic projections in response to photo-evoked PVH glutamatergic input. However, IPSCs from a small portion of neurons (4 out of 12) were partially blocked by TTX/4-AP (Fig. 1j), suggesting a minor portion of monosynaptic GABAergic projection. Supporting this, LSv neurons recorded in *Sim1-Cre:: Vglut2^{flox/flox}* mice, in which synaptic glutamate release is disrupted[36], showed no photo-evoked EPSCs (data not shown) and only a small number of IPSCs (3 out of 25 recorded), which were resistant to TTX/4-AP (Fig. 1k). Consistently, when we delivered to wild type mouse LSv of a mixture of viral vectors containing AAV-DIO-mCherry and retro-AAV vectors AAVrg-Cre-Venus, which can be taken up by synaptic terminals and trace projection neurons[37], a significant number of Venus-only neurons were found away from the injection site with mCherry-expressing neurons (Supplementary Fig. S1l), suggesting Venus-only neurons as local-projecting neurons. To validate the specificity of the connection between PVH and LSv, we ruled out a potential contribution from Sim1 neurons that reside in the nearby posterior hypothalamus (Supplementary Fig. S2). Taken together, these results suggest that local LSv GABAergic circuits receive direct PVH projections consisting of mainly glutamatergic, and a small portion of GABAergic input (Fig. 1l).

**PVH→LSv projections drive negative emotional states.** We next sought to determine what behavioral effects that modulation of

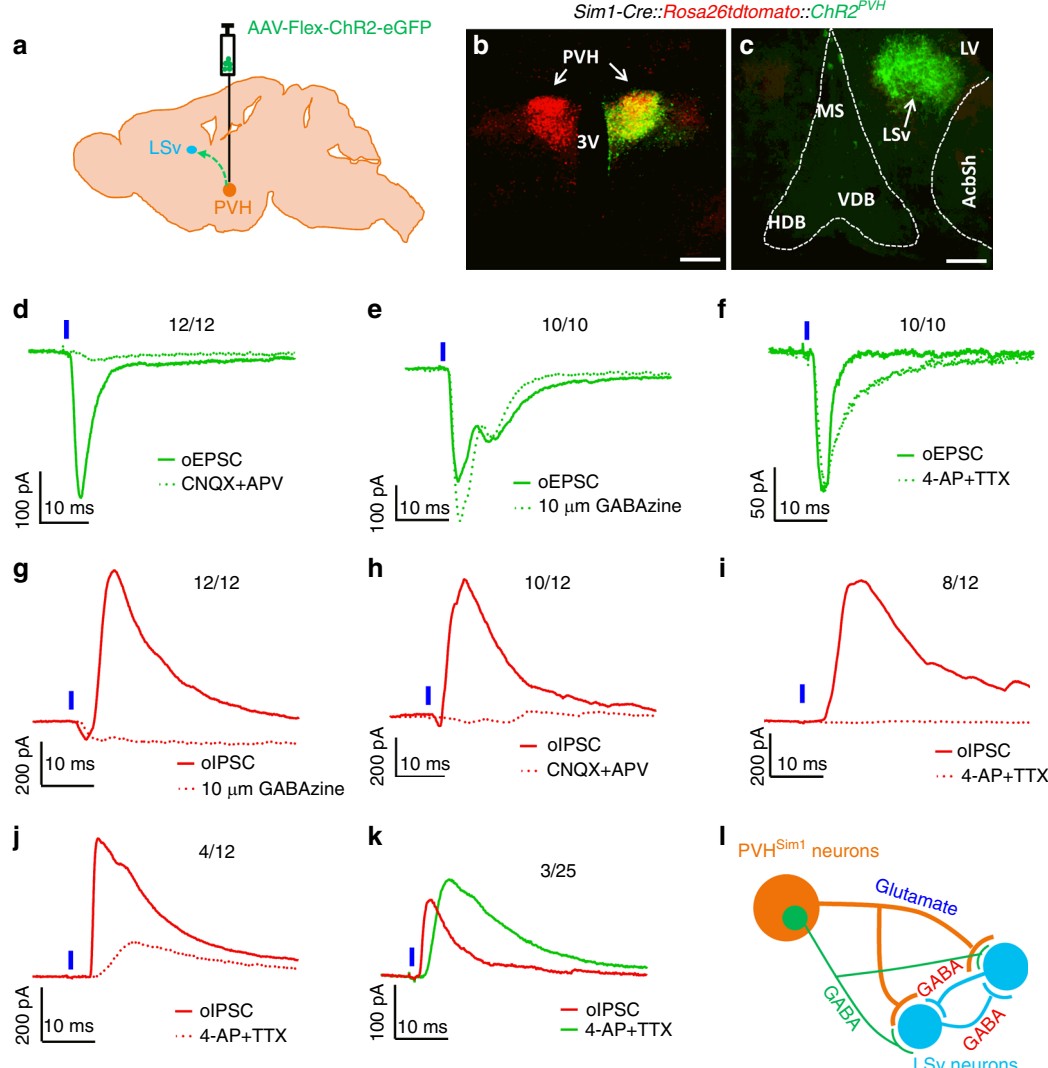

**Fig. 1** A local GABAergic network in the ventral part of lateral septum (LSv) receives direct glutamatergic input from PVH neurons. **a–c** Sim1-Cre reporter mice (four males and two females) received injections of AAV-Flex-ChR2-eGFP vectors to one side of the PVH **a**, facilitating dual expression of the ROSA-lsl-tdTomato allele in Sim1-Cre neurons, and unilateral targeted expression of the AAV-Flex-ChR2-eGFP **b**. **c** GFP-expressing fibers from the PVH shown in the ventral part of lateral septum (LSv) were observed 4 weeks after AAV-delivery. **d–k** Patch clamp recording of randomly selected neurons in LSv in brain slices from Sim1-Cre **d–j** or Sim1-Cre::Vglut2flox/flox mice **k** with photostimulation of PVH→LSv fibers. **d–f** Voltage–clamp recordings for photostimulated (1 ms, blue ticks) excitatory postsynaptic currents (EPSCs) and their responses to glutamate receptor antagonists (CNQX+APV) **d**, GABA-A receptor antagonists (GABAzine) **e**, and 4-AP+TTX to block action potentials **f**. **g–j** Voltage–clamp recordings with photostimulation (1 ms, blue ticks) to monitor inhibitory postsynaptic currents (oIPSCs) and their responses to GABAzine **g**, CNQX+APV **h**, 4-AP+TTX **i** and **j**. **i** and **j** showed the recording from the same set of neurons with complete **i** or partial blockage **j** by 4-AP/TTX. **k** Only three out of 25 recorded neurons showed 4-AP/TTX-resistant IPSC in Sim1-Cre::Vglut2flox/flox mice. The ratios at the top of traces indicate the number of neurons that exhibited responses to drugs out of all neurons showing postsynaptic currents. **l** A diagram derived from the recording data depicting a GABAergic network in the LSv receiving monosynaptic projections from the PVH comprising a major glutamatergic, and minor GABAergic components. Scale bar = 100 μm. PVH: paraventricular hypothalamus; LSv: ventral part of lateral septum; MS; medial septum; HDB and VDB: diagonal band (horizontal and vertical); LV: lateral ventricle; 3 V: the third ventricle; AcbSh: accumbens shell

this circuit may manifest in vivo. Towards this, we implanted fiberoptics over the lateral septum, and stimulated the PVH terminals with blue light (Fig. 2a–c). Guided by our previous studies with different stimulation protocols producing distinct behaviors[33], we used photostimulation with short (10 ms, 5 Hz, 5 mW/mm²) and long light pulses (100 ms, 5 Hz, 5 mW/mm²). First, we confirmed that long pulses of light induced stronger c-Fos expression in the LSv compared with short pulses (Supplementary Fig. S3), suggesting that long pulse stimulation elicits more glutamate release from PVH→LSv terminals, thereby inducing more activation of LSv neurons. Reminiscent of our

previous studies showing that photostimulation of PVH neurons or LH→PVH glutamatergic terminals induces intense stress-related self-grooming[33], unilateral short pulse photostimulation of PVH→LSv terminals similarly induced intense self-grooming behavior (Fig. 2d, Supplementary Movie 1). Analysis of grooming microstructure showed the induced grooming exhibited more incorrect transitions and interrupted bouts (Fig. 2e), suggesting stress-related nature of the grooming behavior[38]. Strikingly, unilateral long pulse photostimulation produced frequent, frantic escape jumping behaviors (Fig. 2f, Supplementary Movie 2). Neither self-grooming, nor jumping was observed in

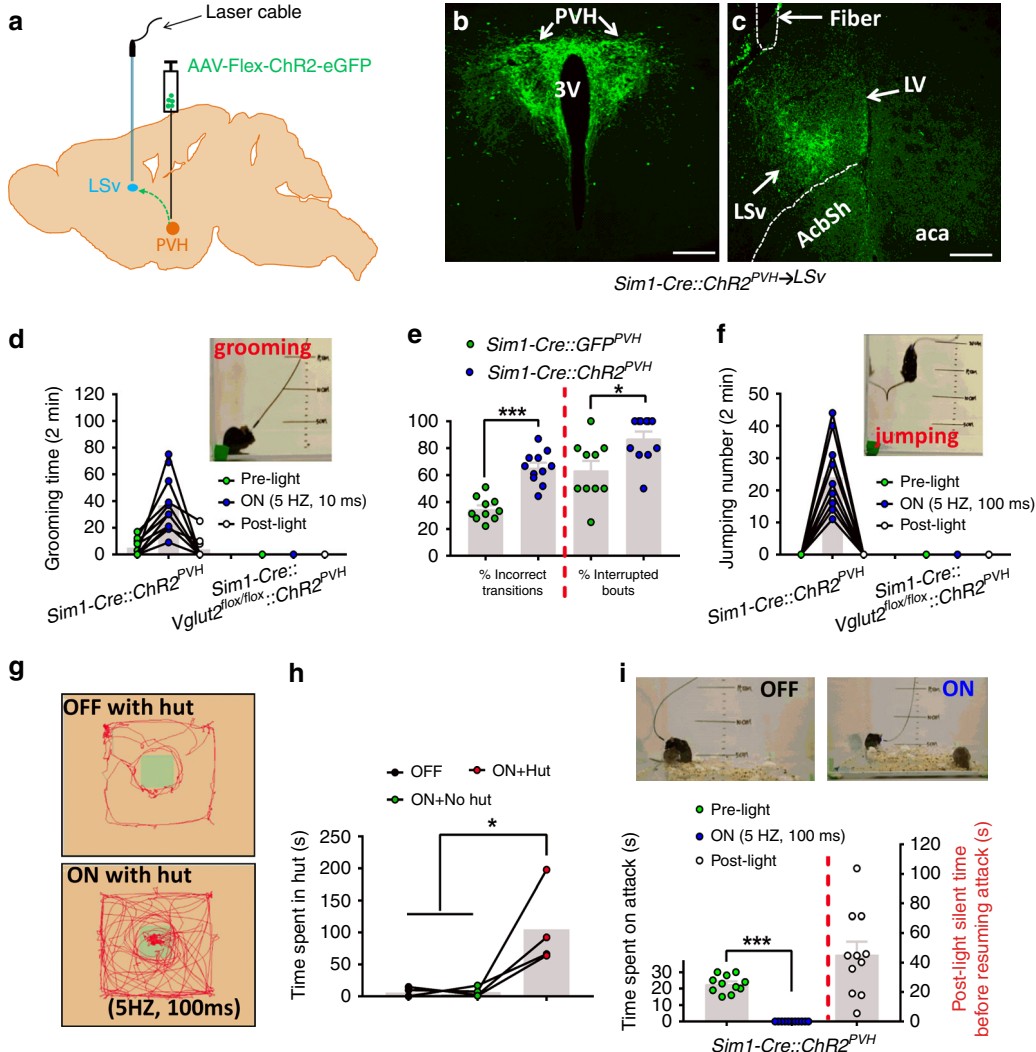

**Fig. 2** Activation of PVH→LSv projections drives behaviors associated with negative emotional states. **a–c** *Sim1-Cre* and *Sim1-Cre::Vglut2flox/flox* mice (males) with AAV-Flex-ChR2-eGFP delivery to the PVH and fiber optic implantation targeting LSv were used for behavioral assays **a**. Representative pictures showing AAV vector expression in the PVH **b**, arrows, and GFP-expressing fibers and optic fiber trace in LSv **c**, arrows. **d–f** These mice were used for a 2 min testing period. In *Sim1-Cre* mice, self-grooming behaviors were induced by short-pulse phototimulation (10 ms/5 Hz, 5 mW/mm²) (**d**, online Movie 1) and frantic jumping behaviors were induced by long pulse photostimulation (100 ms/5 Hz, 5 mW/mm²) (**f**, online Movie 2), neither of which were observed in *Sim1-Cre::Vglut2flox/flox* mice **d** and **f**. **e** Quantitation of self-grooming behaviors shows increased bout and transition (*n* = 10–11; comparison between % incorrect transitions, *p* < 0.001; comparison between % interrupted bouts, *p* = 0.012). **g**, **h** Mice were used for a shelter box test in which mice movements were tracked in an open arena with box (shaded areas in **g**) placed in the center. More time spent in the box represents an increase in fear. Photostimulation of PVH→LSv terminals increased time spent in the box, or in the same area the box occupied, compared with either photostimalation without the box or without photostimulation (**h**, *n* = 4, *p* = 0.0108 or *p* = 0.0113). **i** Representative snapshot pictures of attacking intruders by resident mice with no (OFF) or the long pulse photostimulation (ON) of PVH→LSv fibers (top pictures), and time spent in attack during the 30 s periods of pre-light and light-on (ON, also online Movie 3), and 2 mins post light (**i**, bottom; *n* = 11, *p* < 0001). *\*p* < 0.05, \*\*\**p* < 0.001, paired student's *t* tests for **e** and **I**, One-way AVONA tests for **h**. Scale bar = 100 μM. PVH: paraventricular hypothalamus; LSv: ventral part of lateral septum; 3 V: the third ventricle; aca: anterior commissure area; AcbSh: accumbens shell

*Sim1-Cre::Vglut2flox/flox* mice lacking glutamate release from Sim1-Cre neurons[36] (Fig. 2e, f), suggesting an absolute requirement of glutamate release for the observed behaviors. Given the known role of the LS in fear[34], we next questioned if the escape jumping behavior represents a state of fear. Toward this, we performed a modified shelter hut test, which was previously used to test a fear response[39]. We placed the shelter hut in center instead of corner to assess the ability of animals to actively search for safety for hiding. Unilateral photostimulation of PVH→LSv terminals significantly increased time spent inside a shelter hut placed in the center of the recording arena compared with without photostimulation, or with photostimulation but without hut (Fig. 2g, h),

suggesting a state of fear elicited by selectively photostimulating the PVH→LSv projection. To test whether photostimulation of PVH→LSv terminals reduces aggression, an emotional state consistent with increased fear, we examined aggression in resident mice during resident-intruder assays. For this, we scored resident mice that were engaged in continuous attacks toward the intruder by chasing, mounting, and fighting (Fig. 2i), all of which were completely blocked upon photostimulation, and immediately resumed post stimulation (Fig. 2i, Supplementary Movie 3). These results collectively demonstrate that activation of the PVH→LSv projection drives rapid changes in emotional states ranging from stress to fear, which requires glutamate release.

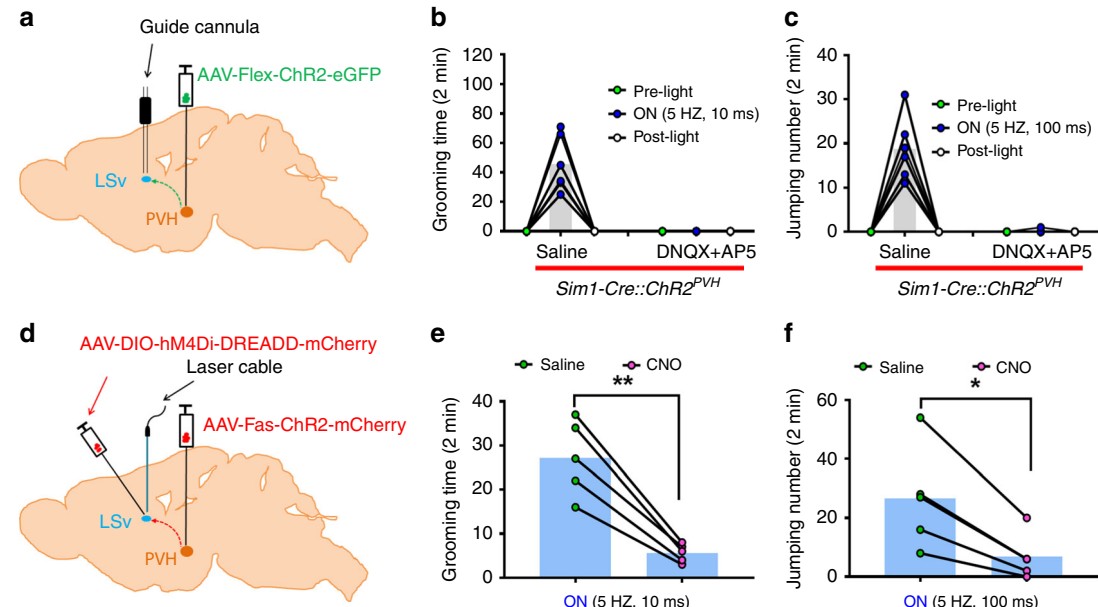

**Fig. 3** LSv neurons mediate the effect of activation of PVH→LSv. **a–c** The effect of blocking LSv glutamate receptors on behavior. A paradigm depicting the experimental strategy using male *Sim1-Cre* mice **a** and the effect on self-grooming **b** and jumping **c** by infusion of saline or glutamate receptor antagonists DNQX+AP5 to LSv 5 mins prior to behavioral testing. **d–f** *Vgat-Cre* male mice with AAV-Fas-ChR2-eGFP ("Cre-off") delivery to PVH, AAV-DIO-hM4Di-EREADD-mCherry delivery to LSv, and optic fiber implantation targeting LSv **d** were used for self-grooming (**e** $n = 5$, $p = 0.0021$) and jumping assays (**f**, $n = 5$, $p = 0.0105$) in response to saline and CNO treatment. PVH: paraventricular hypothalamus; LSv: ventral part of lateral septum; aca: anterior commissure area; 3 V: the third ventricle; AcbSh: accumbens shell. *$p < 0.05$, ** $p < 0.01$, paired student's *t* tests **e**, **f**

## LSv mediates the effect elicited by PVH→LSv projections.

To confirm the role of LSv neurons in mediating the observed behaviors elicited by photostimulation, we infused glutamate receptor antagonists through an implanted cannulae in the LSv (Fig. 3a), as similarly done previously in the PVH[33]. Sim1-Cre with ChR2 expression in the PVH and optic fiber implantation targeting the LSv were used. Both behaviors elicited by both short and long pulse stimulations were effectively blocked by local infusion of AP5/DNQX to the LSv (Fig. 3b, c), confirming that the activation of glutamate receptors in LSv neurons is required for the observed behaviors. Since the vast majority of PVH neurons are glutamatergic[36] and LSv neurons are GABAergic[40], we next delivered AAV-fas-ChR2-mCherry ("Cre-off")[41] to the PVH and AAV-DIO-H4MGi-DREADD-mCherry ("Cre-on") to the LSv of *Vgat-Cre* mice, for targeted activation of PVH neurons and inhibition of LSv neurons, respectively (Fig. 3d). Activation of DREADD by CNO administration was confirmed to reduce c-Fos expression induced by photostimulation of PVH→LSv fibers (Supplementary Fig. S4a–h), and CNO administration also effectively blocked both self-grooming (Fig. 3e) and jumping (Fig. 3f). Notably, CNO administration alone in a separate group of Sim1 mice without DREADD expression in the LSv had no effect on either self-grooming (Supplementary Fig. S4i) or jumping (Supplementary Fig. 4j) induced by photostimulation of PVH→LSv fibers. These results suggest that activation of LSv GABAergic neurons is required for both behaviors.

## PVH→LSv projections elicit negative valence.

Stress-related emotional states are unpleasant, so we next sought to determine whether this circuit node has a more general role in behaviors related to negative emotional states. Indeed, using real time place preference (RTPP) tests, we found that photostimulation of PVH→LSv terminals reduced time spent in the side paired with photostimulation (ON), suggesting aversion to photostimulation (Fig. 4a, b). Compared with control groups, both short (Fig. 4c) and long (Fig. 4d) light pulses produced place avoidance.

Importantly, light pulse stimulation causes a duration-dependent avoidance to the side paired with photostimulation (Fig. 4e), suggesting a scalable regulation via this circuit on negative valence. Fear over-generalization leads to anxiety[42]. Mice normally spend the majority of time in the perimeter during open field tests to avoid anxiety associated with exposure to center areas (Fig. 4f, g). In response to photostimulation, mice with ChR2 expression in *Sim1-Cre* neurons markedly reduced their time spent in the perimeter (Fig. 4f, g). The same photo-illumination in either *Sim1-Cre* mice with control GFP expression, or *Sim1-Cre::Vglut2*$^{flox/flox}$ mice with ChR2-GFP expression, resulted in no alterations in preference (Fig. 4h). Thus, the negative valence elicited by the PVH→LSv activation overcomes anxiety associated with center exposure. Notably, mice with AP5/DNQX infusion to the LSv displayed reduced anxiety, spending more time in open arms in elevated plus maze tests (Fig. 4i) and more time in the center of recording arenas in open-field tests (Supplementary Fig. S5), suggesting that blocking ongoing glutamatergic action in LSv neurons reduces anxiety-like behavior, consistent with an increased anxiety level by photostimulating the PVH→LSv circuit. Taken together, these observations suggest that the PVH→LSv glutamatergic projection drives behaviors related to negative emotional states.

## PVH→LSv projections regulate feeding.

Given the well-established role of PVH in feeding regulation, we examined the role of the PVH→LSv projection in feeding regulation. Strikingly, we found that photostimulation of PVH→LSv terminals with the long light pulse (100 ms, 5 Hz, 5 mW/mm$^2$) markedly reduced refeeding after 20 hr fasting (Fig. 5a), and produced rapid and reversible feeding inhibition to light exposure (Fig. 5b, Supplementary Movie 4). Notably, these effects were not observed in control AAV-Flex-GFP injected *Sim1-Cre* mice (data not shown), or in AAV-Flex-ChR2-eGFP injected *Sim1-Cre::Vglut2*$^{flox/flox}$ mice (Fig. 5a, b), suggesting the necessity of glutamate release from the PVH for the observed inhibitory effect on feeding.

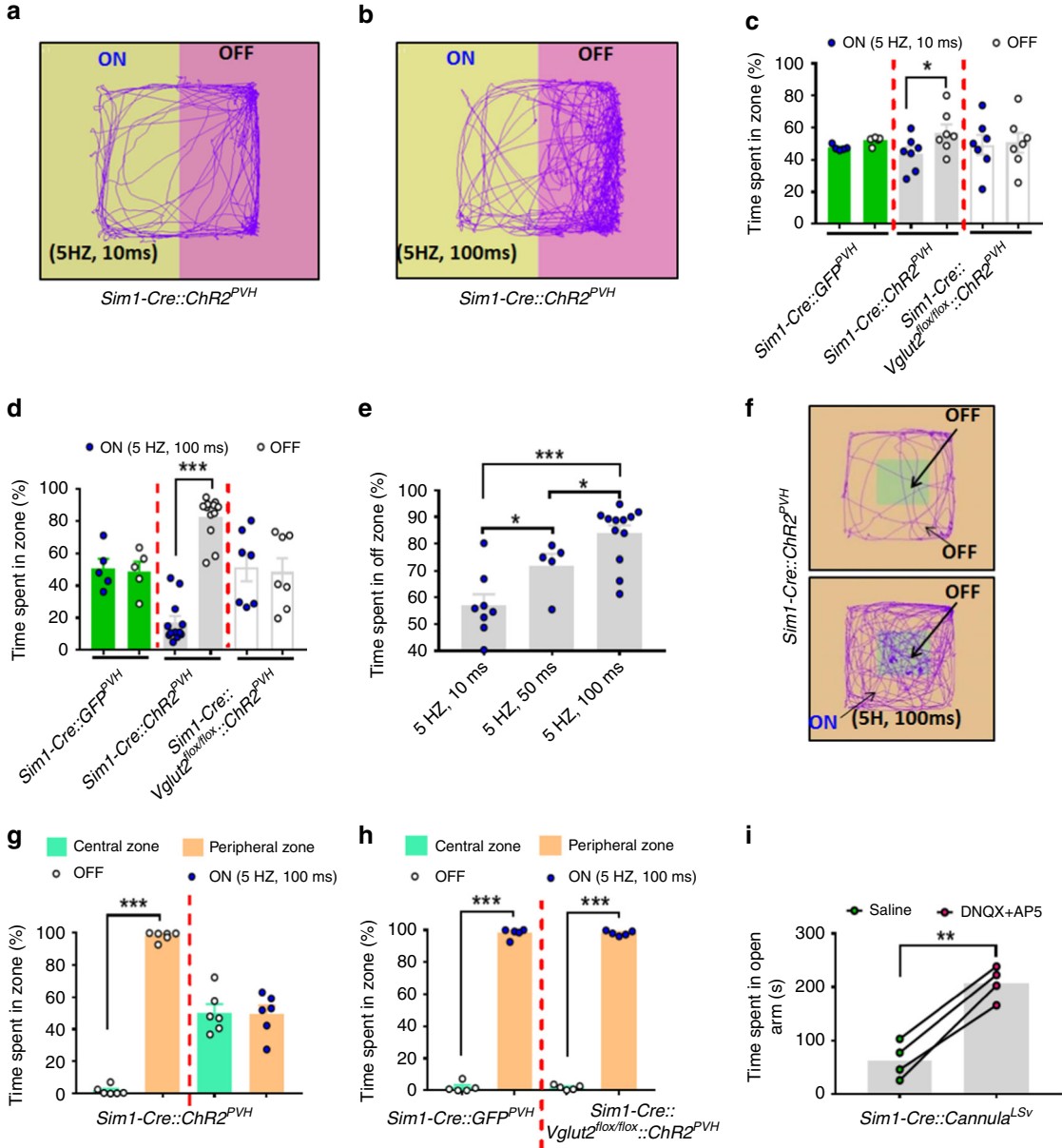

**Fig. 4** Photostimulation of PVH→LSv terminals produces negative valence. **a–e** Real time place preference were conducted in male mice with photostimulation of PVH→LSv fibers. A representative mouse locomotion trace was shown with short **a** and long **b** pulse photostimulation in *Sim1-Cre* mice. Time spent during laser on (ON) and off zones (OFF) in *Sim1-Cre* and *Sim1-Cre::Vglut2^flox/flox* mice with AAV-Flex-ChR2-eGFP delivery, or *Sim1-Cre* mice with AAV-Flex-GFP delivery using the short pulse (**c**, *Sim1-Cre::ChR2^PVH* mice $n = 7$, $p = 0.0321$) and the long pulse stimulation (**d**, *Sim1-Cre::ChR2^PVH* mice $n = 12$, $p < 0.0001$). **e** Time spent in OFF zone shows dependence on the photostimulation duration with fixed 5 Hz and 5 mW/mm² light ($n = 5$; 5 HZ,10 ms vs 5 HZ,50 ms, $p = 0.0452$; 5 HZ,50 ms vs 5 HZ,100 ms, $p = 0.0356$; 5 HZ,10 ms vs 5 HZ,50 ms, $p < 0.0001$). **f–h** Mice described in **a–e** were tested in a modified RTPP, in which photostimulation was paired with the periphery of the arena (**f**, uncolored area). Representative movement traces were shown with no pairing (**f**, upper) or pairing (**f**, bottom), time spent in zones with laser on or off from mice with the indicated genotypes (**g** and **h**, $n = 6$, $p < 0.0001$). **i** Time spent in open arms during elevated plus maze tests in mice with local infusion of glutamate receptor antagonists DNQX/AP5 to the LSv ($n = 4$, $p = 0.0012$). $n = 4$–12. *$p < 0.05$, ***$p < 0.001$, two-way AVONA tests (**c**, **d** and **g**, **h**), One-way AVONA tests **e** or paired student's *t* tests **i**

Interestingly, although photostimulation with long pulses of light (100 ms, 5 Hz, 5 mW/mm²) potently reduced refeeding after 12 hrs or 6 hrs fasting, short pulses of light (10 ms, 5 Hz, 5 mW/mm²) stimulation had no effects on refeeding after 20 hrs of fasting, marginal effects on refeeding after 12 hrs fasting, but significantly reduced refeeding after 6 hrs fasting (Fig. 5a), suggesting a scalable effect on feeding inhibition.

As LSv neurons are mostly GABAergic[40], we also sought to evaluate the function of LSv GABAergic neurons in feeding. Toward this, we photostimulated LSv vesicular GABA transporter

(VGAT)-expressing neurons after delivery of AAV-Flex-ChR2-eGFP vectors to the LSv of *Vgat-Cre* mice (Fig. 5c). Interestingly, photostimulation greatly reduced feeding during a 15 min refeeding period after 12 hr of fasting (Fig. 5d), and produced rapid and reversible inhibition of fast-refeeding that was precisely timed to periods of light exposure (Fig. 5e, Supplementary Movie 5). Consistent with these data, infusion of the glutamate receptor antagonists DNQX/AP5 to the LSv instantly elicited feeding in well-fed mice (Fig. 5f and Supplementary Movie 6), and photoinhibition of PVH→LSv fibers expressing AAV-Flex-eArchT3.0-GFP

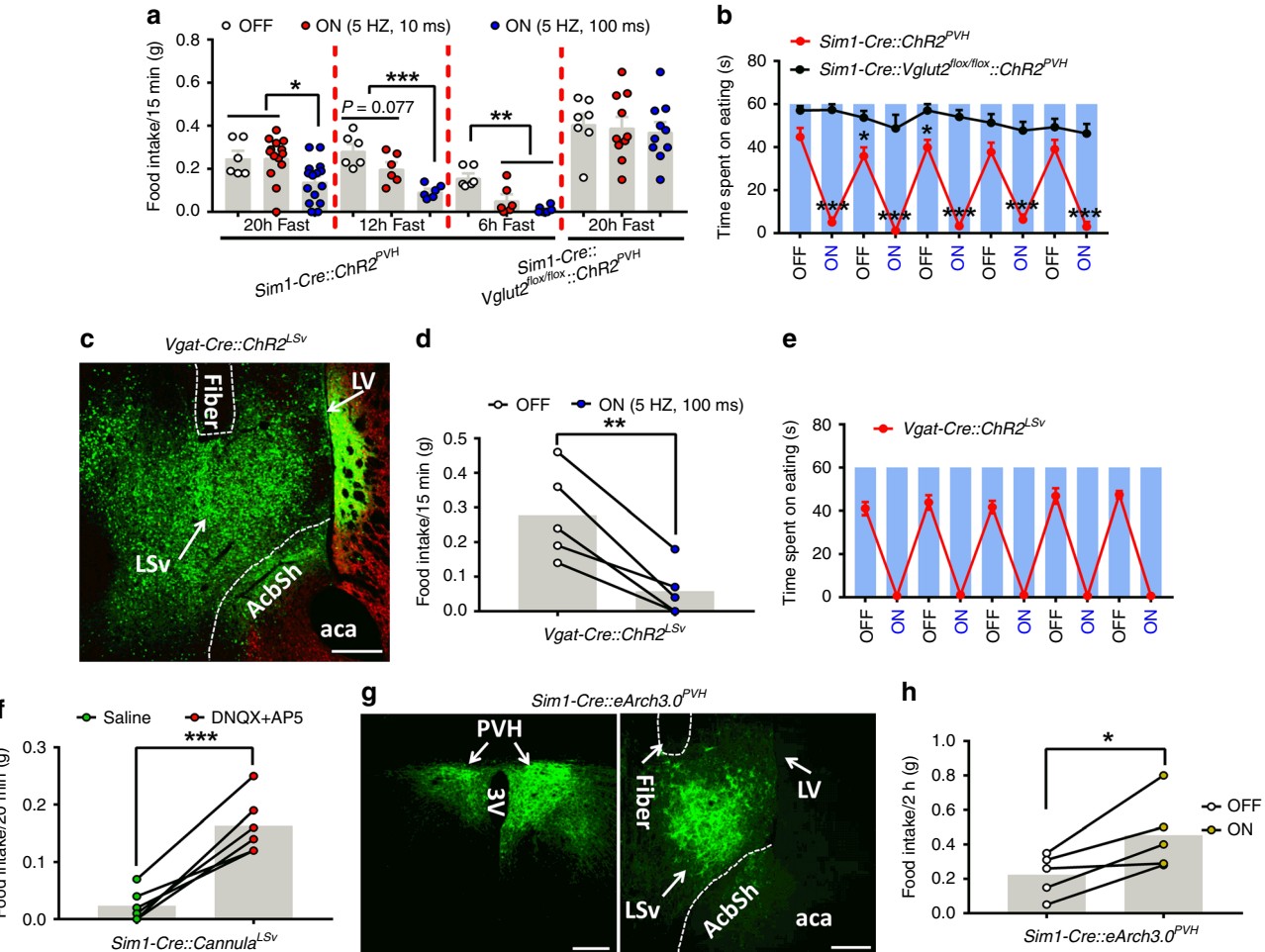

**Fig. 5** PVH→LSv projections regulate feeding. *Sim1-Cre* and *Sim1-Cre::Vglut2flox/flox* male mice with AAV-Flex-ChR2-eGFP delivery to the PVH and fiber optic implantation targeting LSv were used for feeding assays. **a** Measurements of food intake over 15 mins after either 20 hr (n = 6–15, OFF vs 5 HZ,100 ms p = 0.0306; 5 HZ,10 ms vs 5 HZ,100 ms p = 0.0108), 12 hr (n = 6, OFF or 5 HZ,100 ms vs 5 HZ,100 ms, p < 0.0001) or 6 hr (n = 6, p = 0.0002) fasting with no photostimulatoin (OFF), long pulse (100 ms/5 Hz, 5 mW/mm²), or short pulse photostimulation (10 ms/5 Hz, 5 mW/mm²). **b** Measurements of feeding time with alternating segments of laser off (OFF) and on (ON, 100 ms/5 Hz, 5 mW/mm²) at 1 min each (n = 5–8/group, p < 0.001, p = 0.0277, or p = 0.0181; also online Movie 4, Source data are provided as a Source Data file). **c–e** A representative image showing AAV-Flex-ChR2-eGFP expression and optic fiber implantation trace in the LSv of *Vgat-Cre* animals after stereotaxic delivery of the vector to LSv **c**, feeding was measured for 15 mins after 12 hr fasting with no light (OFF), long pulse photostimulation (ON) (**d**, n = 5, p = 0.0049), or measured with alternating segments of laser off (OFF) and the long pulse (ON) at 1 min each (n = 5) (**e**, also online Movie 5, Source data are provided as a Source Data file). **f** Well-fed mice with cannulation targeting LSv were measured for feeding 20 mins after infusion of AP5+DNQX or saline (n = 5, p < 0.0001, also online Movie 6). **g–h** Well-fed *Sim1-Cre* mice with AAV-Flex-ArchT3.0-eGFP bilateral delivery to PVH (**g**, left panel, arrows) and optic fiber implantation targeting LSv harboring GFP-expressing fibers (**g**, right panel, arrows) were measured for feeding in 2 hrs with yellow laser off (OFF) and following a long pulse on (ON, n = 5, p = 0.0268) **h**. *p < 0.05, ***p <0.001, student's t tests for comparison between two groups (**b**, **d**, **f**, and **h**) or ANOVA tests for comparison among three groups **a**. Scale bar = 100 μM. PVH: paraventricular hypothalamus; LV: lateral ventricle; 3 V: the third ventricle; LSv: ventral part of lateral septum; aca: anterior commissure area; AcbSh: accumbens shell. Source Data

increased food intake in well-fed mice (Fig. 5g, h), suggesting a physiological role for tonic glutamate release from the PVH→LSv fibers towards the maintenance of the non-feeding fed state, and that changes in tonic glutamatergic neurotransmission from the PVH to LSv GABAergic neurons dynamically regulate feeding behaviors. Thus, in contrast to previous studies focusing on identification of potential PVH downstream neurons in brain stem and hindbrain[23,24], our results reveal the LSv as a novel downstream site that has a critical role in mediating the PVH action on feeding.

**Stimulating PVH inputs to LSv impairs feeding due to aversion.** Motivation of feeding competes with negative valence

encoding environmental dangers[2,13]. We next used a modified RTPP assay to test competition between hunger-driven feeding and negative valence elicited by stimulation of the PVH→LSv projection. For this, food was made available only in the light-paired side to test refeeding in 12h-fasted Sim1 ChR2-expressing mice (AAV-Flex-ChR2-eGFP injected) compared with controls (AAV-Flex-GFP injected) (Fig. 6a–c) or Sim1 ChR2-expressing *Sim1-Cre::Vglut2flox/flox* mice (Fig. 6d). Whereas control mice exhibited preference to the side with food (Fig. 6a, e) and exhibited normal fast-refeeding (Fig. 6g), long-pulse photo-stimulation of PVH→LSv terminals of *Sim1-Cre* mice resulted in profound avoidance to the paired side (Fig. 6c, e), and consequently showed no feeding behavior (Fig. 6h). Interestingly, short light pulses produced an immediate effect on both avoidance and

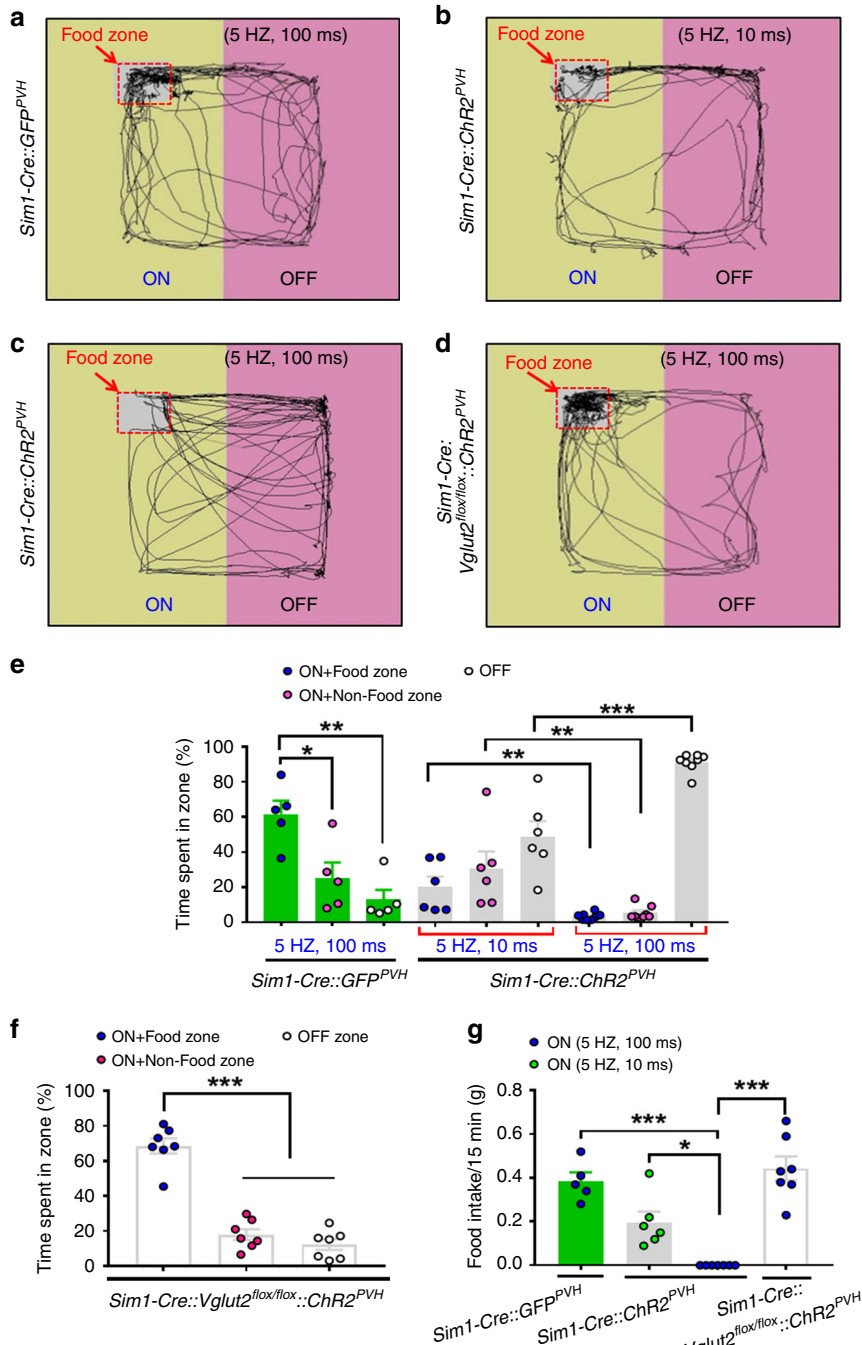

**Fig. 6** Place aversion by photostimulation of PVH→LSv terminals antagonizes hunger-driven feeding. Male mice with AAV-Flex-ChR2-eGFP delivery to PVH and optic fiber implantation targeting LSv were fasted 12 hr before RTPP tests, in which chow diet pellets were placed in the corner of the side paired with photostimulation PVH→LSv fibers **a–d**. Representative movement traces were shown for *Sim1-Cre* control mice with AAV-Flex-GFP injection **a**, *Sim1-Cre* mice with short pulse (ON,10 ms, 5 Hz, 5 mW/mm², **b**, long pulse photostimulation (ON,100 ms, 5 Hz, 5 mW/mm², **c**) and for *Sim1-Cre::Vglut2flox/flox* mice with long pulse photostimulation (ON, **d**). **e–g** Time spent in zones in *Sim1-Cre* mice with AAV-Flex-GFP delivery (**e**, $n = 5$; On+Food zone *vs* On+Non-food zone, $p = 0.0126$; On+Food zone *vs* OFF zone, $p = 0.0016$) or with AAV-Flex-ChR2-eGFP delivery, and *Sim1-Cre::Vglut2flox/flox* mice with AAV-Flex-ChR2-eGFP delivery (**f**, $n = 7$; On+Food zone *vs* On+Non-food zone or OFF zone, $p < 0.0001$). **g** Food intake was measured during a 15 min testing period in **d–f**, $n = 5$–7/group (Sim1-Cre mice 5 HZ,10 ms vs 5 HZ,100 ms, $p = 0.0130$; all other comparisons, $p < 0.0001$). *$p < 0.05$, **$p < 0.01$ and ***$p < 0.001$, one-way (**e**, **f**, and **g**) or two-way (**e** ANOVA tests

feeding (Fig. 6b–g), suggesting an inverse relationship between the level of aversion and hunger-driven feeding. In contrast, the same long pulse photostimulation did not alter food preference or fasting-refeeding responses in *Sim1-Cre::Vglut2flox/flox* mice (Fig. 6d, f, g), reinforcing the role for glutamate release in aversion and feeding inhibition. These results suggest that stimulation of

PVH→LSv terminals reduces feeding by orchestrating behaviors related to negative emotion.

**Identity of PVH neurons that send projections to LSv.** The PVH is composed of diverse and functionally distinct groups of

neurons[28]. For the observed behaviors, we failed to find evidence for a contribution of PVH oxytocin neurons to the PVH→LSv fibers (Supplementary Fig. S6a–c) or for a contribution of these neurons to behaviors (Supplementary Fig. S6d–f). We also had the same observation for the CRH neurons (Supplementary Fig. S7). Similarly, we did not find evidence for a contribution of PVH vasopressin (AVP) neurons to PVH→LSv fibers (Supplementary Fig. S8). However, we noted relatively small amount of GFP-positive fibers in the LSv of CRH receptor 1 (CRHR1)-Cre mice[43] following delivery of AAV-Flex-ChR2-eGFP to the PVH (Supplementary Fig. S9a, b). Photostimulation of PVH^CRHR1→LSv fibers elicited no behavioral changes with short pulse stimulation (data not shown), but induced mild self-grooming and aversion with no significant effects on feeding or jumping with long pulse stimulation (Supplementary Fig. S9). Thus, although PVH^CRHR1 neurons partly contribute to the observed function of PVH→LSv projections, those PVH neurons lacking CRHR1, CRH, oxytocin, and AVP constitute the major component of this behavioral circuit.

**PVH and LSv neurons are sensitive to environmental stressors**. Behavior manifestation related to negative emotion normally responds to environmental cues encoding danger. To examine the function of the PVH→LSv projection in sensing environmental cures, we targeted PVH and LSv neurons for GCaMP6m expression, and monitored their activity using fiber photometry (Supplementary Fig. S10a, b). Both PVH (Fig. 7a, c, e, g) and LSv neurons (Fig. 7b, d, f, h) neuron activities were rapidly increased following a sudden loud sound (Fig. 7a, b), an auditory stressor; light exposure in dark environment (Fig. 7c, d), a visual stressor; and water spray (Fig. 7e, f), an unpleasant touch stressor that produces intense self-grooming. These results suggest an in vivo function of both PVH and LSv neurons in processing environmental stress-producing cues. Conversely, both PVH (Fig. 7g, i) and LSv (Fig. 7h, i) neuron activities showed reduced activity during eating bouts, supporting a role of reducing PVH→LSv neurotransmission in natural feeding behavior.

**Functional tracing between PVH and LSv**. To further define the PVH→LSv projection, we used both retrograde and anterograde tracing approaches to identify both projecting neurons in the PVH and receiving neurons in the LSv. We delivered AAVrg-Cre-Venus vectors to the LSv of adult wild-type mice (Fig. 8a). The AAVrg-Cre-Venus vectors can be traced to presynaptic neurons in a retrograde fashion[37]. The viral vector was delivered to the LSv regions whether intense PVH→LSv fibers were located (Fig. 8b). Numerous Venus-expressing neurons were found in the rostral part, but less was found in the posterior part, of the PVH (Fig. 8c–h), suggesting that the LSv-projecting neurons were predominantly located in rostral PVH. To identify LSv receiving neurons, we delivered AAV1-Cre-GFP viral vectors to the PVH of wild-type mice. The AAV1-Cre-GFP vector can trace immediate downstream neurons in an anterograde fashion[44]. We also delivered a large volume of the AAV-DIO-mCherry vector to the LSv of adult wild-type mice to mark the LSv neurons targeted by the anterograde Cre-expressing vectors (Fig. 8i). The AAV1-Cre-GFP vectors were delivered to the PVH (Fig. 8j) and abundant mCherry-expressing neurons were found in the LSv (Fig. 8k, l). These results further reveal the location of both PVH projecting and LSv receiving neurons of the PVH→LSv projection. Notably, owin to inherent variations associated with the stereotaxic injections, both tracing studies might underestimate the neurons in the PVH and LSv involved in the PVH→LSv projection. Also owin to the same reason, it cannot be excluded that our virus delivery to the PVH may leak into regions surrounding

the PVH, which may implicate non-PVH neuron projection to the LSv. Given our compelling evidence on PVH Sim1 neuron projection to LSv, however, this possibility is very unlikely.

To specifically examine the function of LSv-projecting neurons in behaviors, we delivered retrograde AAVrg-Cre-Venus to the LSv and Cre-dependent AAV-Flex-ChR2-eGFP to the PVH, and implanted optic fibers targeting the LSv for photostimulation of PVH→LSv fibers from PVH neurons traced by the retrograde vectors (Fig. 9a). Three months after surgery, photostimulation with short duration pulses (10 ms, 5 Hz, 5 mW/mm²) induced self-grooming behavior (Fig. 9b) and with long duration pulses (100 ms, 5 Hz, 5 mW/mm²) induced jumping behavior (Fig. 9c). In addition, the short pulse stimulation caused a marginal effect on reducing refeeding of 12 hr fasting, whereas the long pulse produced a dramatic effect on the reduction (Fig. 9d), consistent with a scalable effect on feeding inhibition shown above. Importantly, we also tested responses from LSv-projecting PVH neurons to environmental stressors. We delivered AAVrg-Cre-Venus to the LSv of adult wild-type mice and at the same delivered Cre-dependent AAV-Flex-GCaMP6m or control AAV-Flex-GFP to the PVH with optic fiber implantation targeting the PVH (Fig. 9e). Injections of vector delivery to the LSv and PVH were confirmed (Supplementary Fig. S10c, d). Six weeks after surgery, there mice were used to test responses for either water spray or loud sound as in Fig. 7. Although GFP-expressing neurons had no response to water spray (Fig. 9f) or loud sound (data not shown), GCaMP6m-expressing PVH neurons showed rapid activation by water spray (Fig. 9g) and loud sound (Fig. 9h). Similarly to PVH Sim1 neurons, whereas LSv-projecting PVH neurons with GFP expression showed no changes during eating bouts (Fig. 9i, k), those with GCaMP6m expression exhibited reduced activity during eating bouts (Fig. 9j, k). These observations consolidate the role of LSv-projecting neurons in mediating the action of the PVH→LSv projection in both feeding inhibition and behaviors related to negative emotions.

## Discussion

A timely and safe feeding behavior, which is essential for survival, requires integration of homeostatic hunger drive and emotional states encoding for potential environmental dangers[2]. Recent emerging studies have demonstrated a role for AgRP neurons in altering behaviors related to emotional states[12–15] and conversely, a role for the amygdala, a main brain emotion area, in feeding regulation[2,20,21], suggesting an increasing realization of the importance for changes in valence and related emotional states in feeding regulation[1]. Here, we identified a novel projection from glutamatergic PVH neurons, a key hypothalamic feeding center, to LSv neurons, a known brain region for fear and aggression[34], in orchestrating behaviors related to emotional states for feeding regulation. Activation of glutamatergic PVH→LSv projection produces a scalable effect on negative valence with associated behavioral outputs ranging from stress-related self-grooming to fear-like jumping behaviors, whereas inhibition of the circuit increases feeding with reduced signs of anxiety. Importantly, negative valence elicited by the PVH→LSv projection competes with hunger-driven feeding. Thus, our results reveal a novel, amygdala-independent neurocicuit underlying emotional control of feeding.

Despite extensive studies on mapping out downstream brain sites for PVH neurons in feeding and other homeostatic regulations[22–24,32], to our knowledge, this is the first functional demonstration that the PVH sends a direct functional projection to the LSv, a basal forebrain region. PVH neurons, especially those expressing CRH, have been well-established to initiate adaptive neuroendocrine responses to stress, whereas CRH

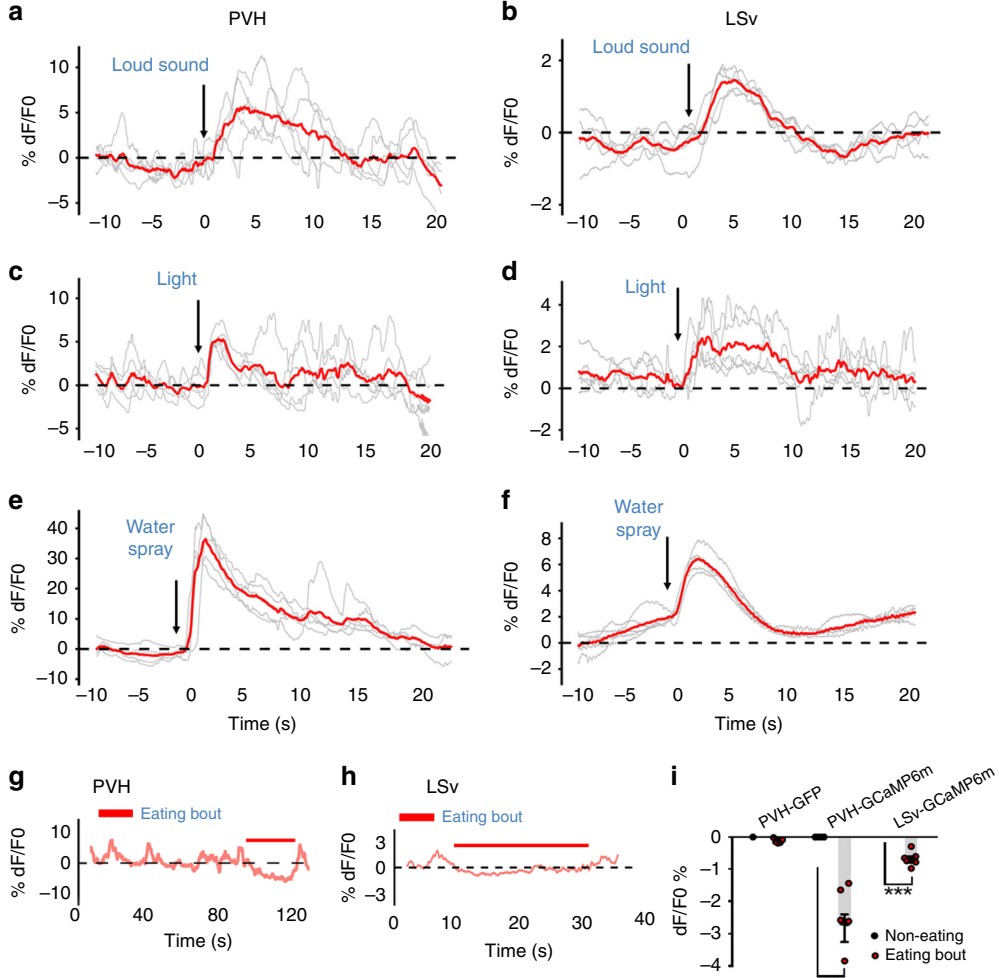

**Fig. 7** In vivo Ca2+ fiber photometry measurements of PVH and LSv neuron activity during stress and feeding. *Sim1-Cre* **a**, **c**, **e**, and **g** and *Vgat-Cre* male mice (**b**, **d**, **f**, and **h**) with AAV-Flex-GCaMP6m delivery to the PVH (**a**, **c**, **e**, and **g**) or LSv (**b**, **d**, **f**, and **h**) were implanted with fiberoptics targeting the PVH or LSv for fiber photometry monitoring the in vivo activity of PVH and LSv neurons in freely moving mice. A loud sound (**a** and **b**), a brief light exposure in dark (**c** and **d**), water spray toward head (**e** and **f**) were associated with rapid activation of PVH (**a**, **c**, **e**, and **g**) and LSv (**b**, **d**, **f**, and **h**) neurons. In contrast, the activity of PVH **g** and LSv **h** neurons was highly associated with eating bouts; and **i** summary data showing comparison in activity changes indicated by averaged means in Ca2+ imaging between periods with feeding bouts and the testing period in the groups indicated. PVH-GFP: recording from PVH neurons expressing GFP mediated by *Sim1-Cre*; PVH-GCaMP6m: recording from PVH neurons expressing GCaMP6m mediated by *Sim1-Cre*, and LSv-GCaMP: recording from LSv neurons expressing GCaMP6m mediated by *Vgat-Cre* (*n* = 8–10 each group). ***$p < 0.001$, two-way ANOVA tests, eating versus non-eating bouts. The red lines in **a**–**f** represent averaged means of traces from the indicated individual stressing events

neurons in the extended amygdala mediate autonomic and behavioral responses to stress[31]. Although PVH CRH neuron projection to the lateral hypothalamus can modulate adaptive behavioral responses after stress[45], a role for PVH neurons has not been postulated to drive behaviors related to fear and anxiety. Our results demonstrate that the PVH→LSv projection directly drives self-grooming, a behavioral manifestation of stress, and escape jumping, a behavioral manifestation of fear. The fear state is also supported by more time spent in the shelter hut and disrupted aggressive behaviors upon photostimulation of PVH→LSv terminals. Thus, PVH neurons are capable of driving a range of behavioral manifestation related to negative emotional states including aversion, fear and anxiety. The in vivo activity of LSv, PVH neurons, and LSv-projecting PVH neurons, is sensitive to and increased by environmental cues encoding for potential danger. Given the demonstrated profound effects on suppressing hunger-driven feeding by competing negative remotional states, the PVH→LSv projection may play an important role in processing and integrating potential environmental risks, whereby

eating can be stopped in a timely manner to avoid predation for survival. Therefore, the identified PVH→LSv projection may represent an evolutionarily conserved brain mechanism for a timely and safe feeding behavior.

One striking finding is that the PVH→LSv circuit exhibits a scalable control over feeding and negative emotional states. The scalable regulation is reflected in an increased feeding by inhibition of PVH→LSv projections, stress-related self-grooming by short pulse stimulation, fear-like escaping jumping behavior by long pulse stimulation and a light pulse duration-dependent effect on aversion. In contrary, inhibition of the PVH→LSv circuit reduced signs of anxiety. Thus, the PVH→LSv projection orchestrates emotional states ranging from a low anxiety level compatible for feeding to fear and anxiety, thereby generating a diverse behavioral repertoire that spans from feeding to escape jumping, suggesting a role of dynamic glutamatergic PVH→LSv neurotransmission in gating emotional states for promoting or inhibiting feeding. It is well-established that inhibition of PVH neurons increases feeding[22,24]. Our data show that specific

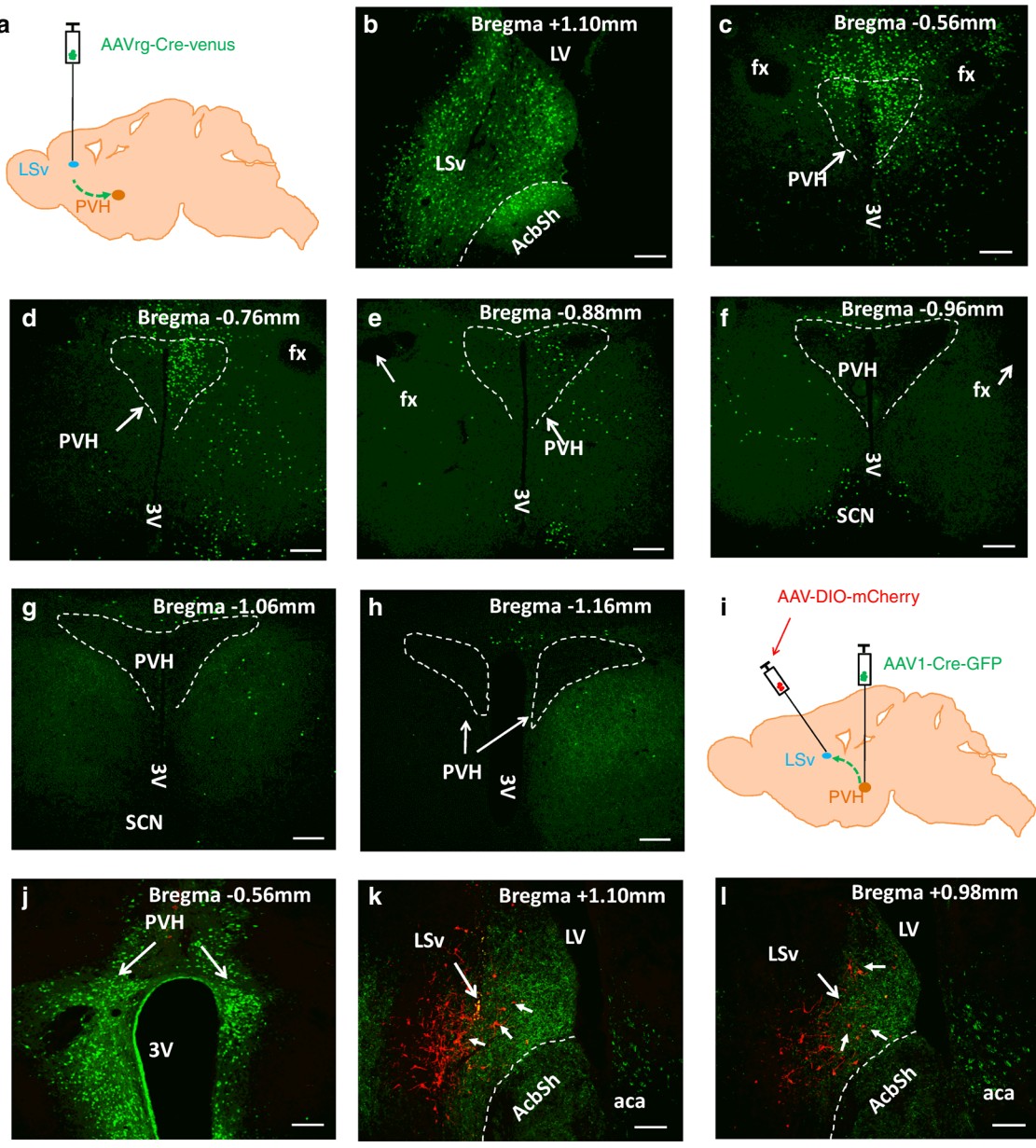

**Fig. 8** Tracing between PVH and LSv. **a–h** Retrograde tracing from the LSv to PVH. **a** Diagram showing delivery of retrograde AAVrg-Cre-Venus to the LSv of adult wild-type male mice and examination of traced upstream neurons including the PVH. Confirmation of vector expression in the LSv **b**, and the traced neurons in brain sections with various Bregma levels from rostral to caudal PVH **c**–**h**. **i**–**l** Anterograde tracing from the PVH to LSv. **i** Diagram showing delivery of anterograde AAV1-Cre-GFP to the PVH of adult wild-type mice and of Cre-dependent AAV-DIO-mCherry vectors to the LSv for examination of traced downstream LSv neurons. Confirmation of AAV1-Cre-GFP expression in the PVH **j** and mCherry-expressing neurons in the LSv **k** and **l**. Scale bar = 100 μM. The white line enclosed areas in **c**–**h** outlining the PVH. Short arrows in **k** and **l** pointing to mCherry-expressing neurons. SCN: suprachiasmatic nucleus; LV: lateral ventricle; LSv: ventral part of lateral septum; fx: fornix; 3 V: the third ventricle; aca: anterior commissure area; AcbSh: accumbens shell

inhibition of PVH→LSv terminals increased feeding and antagonism of glutamate receptors in the LSv elicited instant feeding in well-fed mice. These observations collectively suggest that the profound feeding inhibition following photostimulation of the PVH→LSv projection is not a passive effect of negative valence but rather a novel fundamental mechanism for feeding regulation. Thus, LSv neurons represent a previously unappreciated downstream brain site in mediating the PVH action on feeding regulation and tonic glutamatergic neurotransmission from the PVH to the LSv has an important role in the maintenance of low feeding level "fed" states.

The mechanism underlying distinct self-grooming and jumping behaviors elicited by respective short and long pulse stimulation is not clear, but may be mediated by a shared PVH→LSv projection or separately mediated by distinct PVH→LSv projections. Notably, in all mice tested, although self-grooming was elicited with weak (short pulse) stimulation and jumping was elicited with strong (long pulse) stimulation, both behaviors can be elicited with an intermediate stimulation with automatic transition between them. This is also true in those mice with stimulation of PVH→LSv fibers from selective LSv-protecting PVH neurons, which involved a much less number of PVH

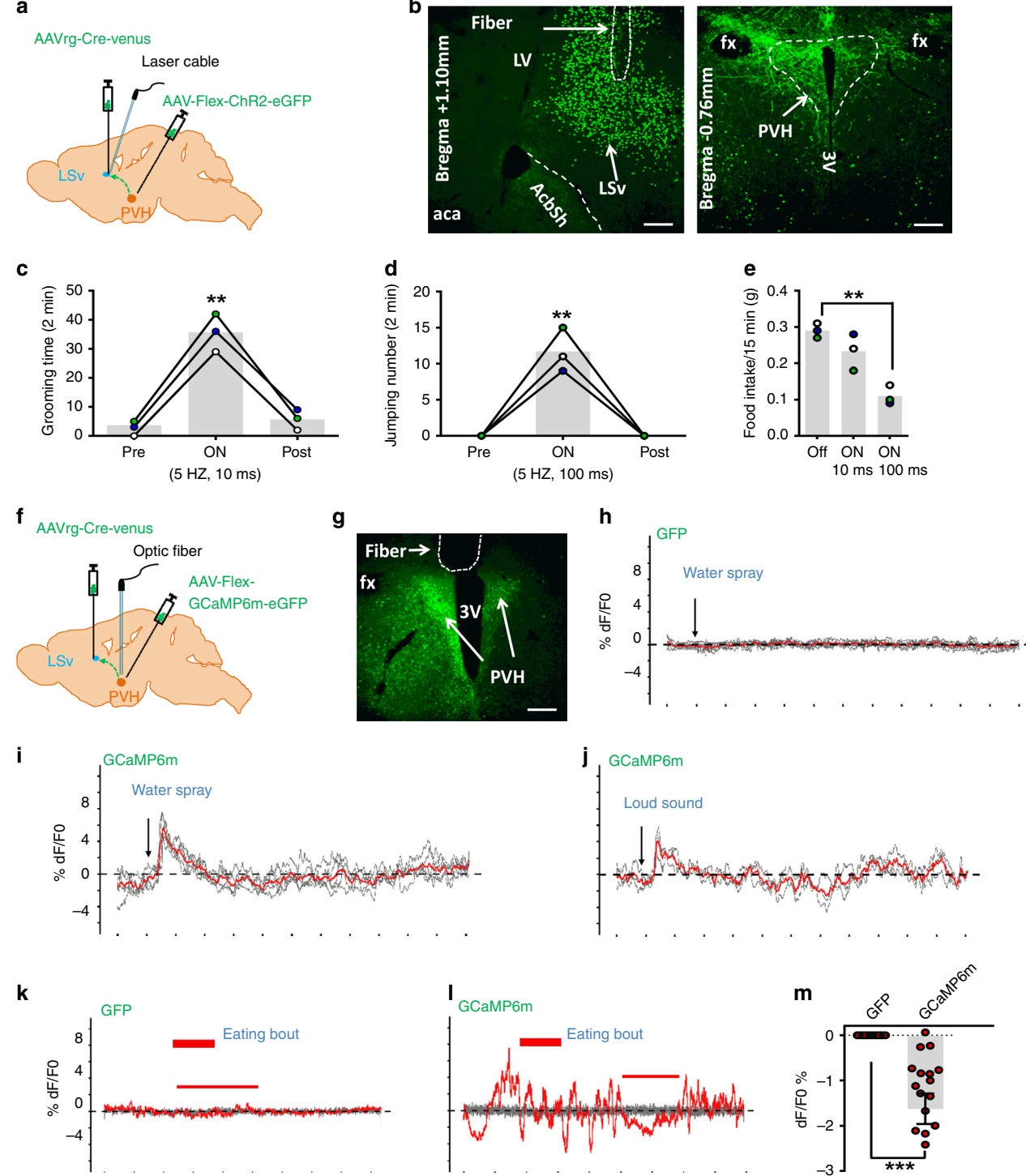

neurons, supporting that the behaviors are mediated by a shared projection. It is conceivable that long pulse stimulation may recruit more ChR2-expressing fibers and therefore trigger more glutamate release, as supported by more c-Fos expression in the LSV with long pulse stimulation. These behaviors are maybe phenotypic manifestations of different levels of negative valence elicited by various strengths of the photostimulation, as suggested previously for other behaviors[46]. Further studies are warranted to discern these possibilities.

Of note, a similar self-grooming behavior was elicited by photostimulation of PVH neurons, or glutamatergic projections from LH to PVH[33], providing evidence for a long range glutamatergic

LH→PVH→LSv projection in promoting stress-related self-grooming behavior. Thus, LH glutamatergic inputs may provide an important excitatory action on the PVH→LSv projection in promoting fear-related negative valence. Consistently, activation of LH glutamatergic neurons causes negative valence[17]. Interestingly, activation of AgRP neurons promotes feeding, while reducing behaviors related to fear and stress[12–15], raising a possibility that the PVH→LSv projection is located downstream of AgRP neurons in mediating feeding and emotional valence. Given the PVH as a major integration center in the brain, it is conceivable that many other brain sites regulate feeding and valence-related emotion through the PVH→LSv projection.

**Fig. 9** Functions of LSv-projecting PVH neurons. **a–d** Behaviors induced by photostimulation of PVH→LSv fibers from selective LSv-projecting neurons. **a** Diagram showing delivery of retrograde AAVrg-Cre-Venus vectors to the LSv (**b**, left panel), and Cre-dependent AAV-Flex-ChR2-eGFP vectors to the PVH (**b**, right panel), and implantation of optic fibers targeting the LSv of adult wild-type mice. Short pulse (10 ms, 5 Hz, 5 mW mm$^2$) photostimulation of PVH→LSv fibers from selective LSv-projecting neurons induced self-grooming (**c**, $n = 3$; light on vs pre-light $p = 0.0003$, light on vs post-light $p = 0.0005$) and long pulse (100 ms, 5 Hz, 5 mW/mm$^2$) induced jumping (**d**, $n = 3$; light on vs pre-light $p = 0.0005$, light on vs post-light $p = 0.0005$); and while the short pulse stimulation caused marginal reduction, the long pulse caused dramatic reduction, on refeeding after 12 hr fasting (**e**, $n = 3$; Off vs 5 HZ, 100 ms $p = 0.0018$). **\*\*$P < 0.01$, one-way ANOVA tests. **f–m** Responses in the activity of LSv-projecting PVH neurons to water spray and loud sound. **e** Diagram showing delivery of retrograde AAVrg-Cre-Venus vectors to the LSv, and Cre-dependent AAV-Flex-GCaMP6m or AAV-Flex-GFP **g** vectors to the PVH, and implantation of optic fibers targeting the PVH of adult wild-type mice. Response to water spray in control mice with GFP expression in traced LSv-projecting neurons **h** and in mice with GCaMP6m expression in those neurons **i**; responses to loud sound in mice with GCaMP6m expression in traced LSv-projecting neurons **j**. **k–m** Representative traces showing changes in Ca2+ signal during a period with feeding bouts in mice with GFP expression **k** or GCaMP6m expression **l** in LSv-projecting neurons. **m** Summary of changes in Ca2+ signal in GFP and GCaMP6m mice. Mice used in feeding were fasted to increase the number of eating bouts ($n = 20$ each, \*\*\*$p < 0.001$, unpaired Student's $t$ tests). The red trace in **h–j** indicating averages of individual traces shown in gray. The red traces in **k** and **l** showing Ca2+-dependent signals, whereas the gray traces showing Ca2+-independent signals. Scale bar = 100 μM. The white line enclosed areas in **b** (right panel) outlining the PVH. LV: lateral ventricle; LSv: ventral part of lateral septum; fx: fornix; 3 V: the third ventricle; aca: anterior commissure area; AcbSh: accumbens shell

Surprisingly, within the PVH, we failed to find evidence for a contribution from CRH, oxytocin, or AVP neurons. Supporting this, oxytocin action in the LS on aggression originates from the supraoptic nucleus[47] and PVH CRH neurons mainly initiate the adaptive neuroendocrine responses, but not feeding responses to stress[30]. Given that TRH neurons within the PVH promote feeding[28], it is unlikely that these neurons contribute to the function of the PVH→LSv projection. Although our data suggest a minor contribution from PVH neurons expressing CRHR1, the identity of the main contributing PVH neurons, which may represent a group of neurons that are distinct from the known PVH subtypes, is not yet clear. Given a major role for PVH neurons that express melanocortin receptor 4 in feeding suppression[23] or glucagon like peptide receptor 1[48,49], these neurons may contribute significantly to the PVH→LSv projection. Importantly, our retrograde tracing results showed that the LSV-projecting PVH neurons are mainly located in the rostral part of the PVH, suggesting a novel type of PVH neurons, which may not be coded by the existing known expression signatures as none of those showed a similar expression pattern. As spinal cord- and hindbrain-projecting neurons are mainly located in caudal PVH[29], it is less likely that the LSv-projecting neurons also project to hindbrain and spinal cord. For downstream targets of this input, we confirmed that GABAergic neurons in the LS mediated both feeding and negative valance by photostimulation of PVH→LSv terminals and identified the location of LSv neurons through anterograde tracing. Given the inherent variation associated with tracing studies, the number of the traced LSv neurons may be underestimated. Further studies using an unbiased approach will be required to reveal the complete subset of LSv neurons that receive direct synaptic inputs from PVH neurons. Nonetheless, LS neurons expressing CRHR2 unlikely serve as a direct target of the PVH→LSv projection, as direct stimulation of these neurons instead promoted aggression[40].

Our results show that glutamate release is required for the evoked feeding and aversion phenotypes described here, pointing to an insufficient role for GABA or PVH neuropeptides within the PVH→LSv projection towards mediating these behaviors. Notably, this is in contrast to previous reports highlighting the importance of PVH peptides in feeding-related behaviors[45,50,51]. Moreover, CRH, oxytocin, and AVP neurons within the PVH do not provide functional contributions to the PVH→LSv projection, suggesting that the PVH→LSv projection may not involve the known PVH peptidergic neurotransmission. Inactivation of glutamate receptors in LSv promoted feeding in the absence of hunger, and reduced baseline anxiety, reminiscent of behavioral states elicited by anti-psychotic drugs[52,53] and/or those observed

in diet-induced obesity models[54]. Activation of the PVH→LSv projection led to overwhelming aversion that was sufficient to overcome hunger-driven feeding, which may relate to behavioral phenotypes in human eating disorders[8,9]. Thus, our findings establish the PVH→LSv projection as a novel neural basis for feeding control by gating stress-related emotional states, and may underlie common circuit features that link behavioral manifestations between psychiatric disorders and eating abnormalities.

## Methods

**Animal care.** Mice were housed at 21 °C–22 °C with a 12 h light/12 h dark cycle with standard pellet chow and water provided ad libitum unless otherwise noted for fasting experiments. Animal care and procedures were approved by the University of Texas Health Science Center at Houston Institutional Animal Care and Use Committee. *Sim1-Cre* and *Sim1-Cre::Vglut2flox/flox* mice were described previously[35,55]. *Oxytocin-Cre, CRH-Cre*, and *AVP-Cre* were obtained from the Jax lab and were described previously[28,56]. *CRHR1-Cre* mice were also described previously[57]. In addition, in most breeding pairs, either male or female breeders (or both) contained the Ai9 reporter gene (Rosa-LSL-tdTomato) to allow RFP expression in the presence of cre recombination[58]. Both male and female mice were used as study subjects, except otherwise noted. All mice that were used for stereotaxic injections were at least 8–10 weeks old.

**Surgeries and viral constructs.** Stereotaxic surgeries to deliver viral constructs and for optical fiber implantation were performed as previously described[59]. In brief, mice were anesthetized with a ketamine/xylazine cocktail (100 mg/kg and 10 mg/kg, respectively), and their heads affixed to a stereotaxic apparatus. Viral vectors were delivered through a 0.5 μL syringe (Neuros Model 7000.5 KH, point style 3; Hamilton, Reno, NV, USA) mounted on a motorized stereotaxic injector (Quintessential Stereotaxic Injector; Stoelting, Wood Dale, IL, USA) at a rate of 30 nL/min. Viral preparations were titered at ~10$^{12}$ particles/mL. Viral delivery was targeted to the PVH (150 nL/side; anteroposterior (AP): − 0.9 mm; mediolateral (ML): ± 0.4 mm; dorsoventral (DV): −4.8 mm) or LSv (200 nL/side; AP: +1.1 mm; ML: ±0.5 mm; DV: − 3.5 mm). Uncleaved fiber optic cannulae (Ø1.25 mm Stainless Ferrule, Ø200 μm Core, 0.39 NA; ThorLabs, Newton, New Jersey, USA) were precut to 3.0 mm and implanted above the LSv.

For blue light-dependent activation of PVH→LSv fibers, *AAV-EF1α-FLEX-hChR2(H134R)-EGFP-WPRE-hGHpA* (Addgene, plasmid number 20298) serotype 2/9[60] was injected unilaterally or bilaterally into the PVH of *Sim1-Cre* and *Sim1-Cre::Vglut2flox/flox* mice, or to the LSv of *Vgat-Cre* mice. For light-dependent inhibition of PVH→LSv fibers, AAV-FLEX-eArchT3.0-eGFP was delivered bilaterally into PVH of *Sim1-Cre* mice. AAV-FLEX-GFP was injected into the PVH of *Sim1-Cre* mice and used as an opsin-negative control for behavioral experiments. AAV-fas-ChR2-mCherry construct[41] was purchased from the Addgene and the virus was prepared in the Baylor NeuroConnectivity Core with a titer at ~10$^{13}$ particles/mL. This expression of ChR2-mCherry will be inactivated by Cre and will therefore be expressed in Cre-negative cells[41]. AAVrg-Cre-Venus (AAVrg-EF1a- iCre-P2A-H2B::Venus) and AAV-DIO-mCherry were both provided from the Baylor NeuroConnectivity Core and injected to the ventral part of lateral septum of wild-type mice. AAV1-Cre-GFP (AAV1-EF1a-EGFP-2A-iCre) viruses were provided from the Baylor NeuroConnectivity Core and injected to the PVH of wild-type mice. AAV-Flex-GCaMP6m (AAV-EF1a-Flex-GCaMP6m) were also provided from the Baylor NeuroConnectivity Core and injected to either PVH or LSv of wild type, *Sim1-Cre*, or *Vgat-Cre* mice. AAV-DIO-DREADD (AAV5-hsyn-DIO-hm4D(Gi)-mcherry) viruses were purchased from the UNC viral core

facility and injected to the LSvs of Vgat-Cre mice. Behavioral testing was conducted during the light cycle following a minimum 4 week recovery period post surgery.

**Brain slice electrophysiological recordings**. Coronal brain slices (250–300 μm) containing the LSv from mice that had received stereotaxic injections of AAV-FLEX-ChR2-eGFP to PVH at least 3 weeks prior to the recordings were cut in ice-cold artificial cerebrospinal fluid (aCSF) containing the following (in mM): 125 NaCl, 2.5 KCl, 1 MgCl₂, 2 CaCl₂, 1.25 NaH₂PO₄, 25 NaHCO₃, and 11 D-glucose bubbling with 95% O₂/5% CO₂. Slices were immediately transferred to a holding chamber and submerged in oxygenated aCSF. Slices were maintained for recovery for at least 1 h at 32–34 °C before transferring to a recording chamber. Individual slices were transferred to a recording chamber mounted on an upright microscope (Olympus BX51WI) and continuously superfused (2 ml/min) with ACSF warmed to 32–34 °C by passing it through a feedback-controlled in-line heater (TC-324B; Warner Instruments). Cells were visualized through a × 40 water–immersion objective with differential interference contrast optics and infrared illumination. Whole cell voltage–clamp recordings were made from neurons within the LSvs that showed the highest density of ChR2-eYFP+ axonal fibers. Patch pipettes (3–5 MΩ) were filled with a Cs⁺-based low Cl⁻ internal solution containing (in mM) 135 CsMeSO₃, 10 HEPES, 1 EGTA, 3.3 QX-314, 4 Mg-ATP, 0.3 Na₂-GTP, 8 Na₂-Phosphocreatine (pH 7.3 adjusted with CsOH; 295 mOsm) for voltage–clamp. For current–clamp recordings, pipettes were filled with a K⁺-based low Cl⁻ internal solution containing (in mM) 145 KGlu, 10 HEPES, 0.2 EGTA, 1 MgCl₂,4 Mg-ATP, 0.3 Na₂-GTP, 10 Na₂-Phosphocreatine (pH 7.3 adjusted with KOH; 295 mOsm). Membrane potentials were corrected for ~10 mV liquid junction potential. To activate ChR2 light from a 473 nm laser (Opto Engine LLC, Midvale, UT, USA) was focused on the area of the recorded LSv neurons to produce spot illumination through optic fiber. Brief pulses of light (blue light (473 nm), 1–2 mW/mm²) were delivered at the recording site at 15 s or 200 ms intervals under control of the acquisition software. GABAzine (10 μM) or CNQX + APV (20 μM and 50 μM) drugs (Abcam) were bath-applied to block GABA-A receptors or AMPA, kainate, and NMDA receptors, respectively, during voltage–clamp recordings of photostimulation-induced inhibitory or excitatory current responses. TTX (0.5 μM; Alomone labs, Jerusalem, Israel), and 4-AP (100 μM; ACROS Organics, Fisher Scientific, Pittsburgh, PA, USA) were bath-applied during voltage–clamp recordings of photostimulation-induced inhibitory and excitatory current responses to block action potentials and inhibit network activity.

**Feeding assays**. For in vivo experiments, an integrated rotary joint patch cable connected the ferrule end of optic fiber cannula with a Ø1.25 mm ferrule end of the patch cable via a mating ceramic sleeve (ThorLabs, Newton, New Jersey, USA). At the other end of the rotary joint, an FC/PC connector was connected to a 473 nm diode-pumped solid state laser (Opto Engine LLC, Midvale, Utah, USA). Light pulses were controlled by Master-8 pulse stimulator (A.M.P.I., Jerusalem, Israel). For optogenetic stimulation-feeding experiments, mice were ad libitum fed prior to testing (excluding competition experiments; see below). Mice after 20 hr or 6 hr fasting underwent 15-min trials of refeeding with laser on or off. During the light-on period, blue light (473 nm, ~ 5 mW/mm²) was pulsed at 5 Hz, with each pulse-width duration lasting 10 or 100 ms. Food intake was measured and recorded after the completion of each epoch during the 15-min trial. Mice after 20 hr fasting were also tested for feeding during alternating epoch of laser on and off. A video was recorded for each trial. An observer blind to the experimental conditions watched the videos and manually calculated the time spent in feeding with a stopwatch. Feeding time was counted when mice were actively engaged in biting, chewing, swallowing, or licking food.

**Grooming and jumping assays**. An observer blind to the genotype condition watched the 6-min videos and manually quantified the time spent grooming during each epoch (2 min each) with a stop watch. Grooming time was counted when the mouse engaged in paw strokes made near the nose, eyes, and head and during paw, body, tail, and genital licking. To assess whether grooming induced by light activation results in abnormal patterning, an observer watched 5 Hz, 10 ms videos of mice and quantified the number of grooming bouts, bout interruptions, and changes in grooming transitions using a grooming analysis algorithm described[61] during pre-light and light-on (5 Hz, 10 ms) conditions. Jumping was defined as the whole body of mice, including all four limbs, being off the floor. Times of jumping were counted within the set 6 min test duration. Time spent in grooming and jumping times were averaged for each epoch (pre-light, light-on and post-light) and compared between epochs.

**Ionotropic GluR blockade experiment**. Custom-made guiding tube cannula with peek M3 receptacle (Doric lenses, Quebec, Canada) allowing for interchangeable fluid and light delivery were implanted above LSv in Sim1-Cre mice containing ChR2 in PVH^Sim1 neurons. Prior to testing, the M3 receptacle was removed from the guiding cannula and the fluid-delivery cannula was inserted via a screw-on top mechanism. Either 100 nL vehicle (15% sterile dimethylsulfoxide; DMSO in 0.9% saline) or 100 nL drugs (Tocris) containing 50 nL (61 mM D-AP5 solution in saline) + 50 nL (24 mM DNQX solution in 25–30% DMSO) were delivered via syringe (5 μL, Model 75 RN SYR, Small Removable NDL, 26s ga, 2 in, point style 2; Hamilton, Reno, NV, USA).

A plastic tube (RenaSil Silicone Rubber Tubing, .025 OD X .012 ID; Braintree Scientific, INC, Braintree, MA, USA) joined the syringe to the fluid-delivery cannula, and vehicle or drugs were manually infused at a rate of 100 nL/min. To prevent backflow of fluid, the fluid-delivery cannula was left screwed on for an additional 2–3 min after infusion. Thereafter, the fluid-delivery cannula was screwed-off and the optical fiber cannula was screwed on and attached to a fiber optic patch cable for light delivery and subsequent behavioral testing. Light was pulsed at 5 Hz, 10 ms, or 100 ms (473 nm, ~ 5 mW/mm²) for 2 mins in each trial, in which a video was recorded and later quantified for time spent grooming and jumping.

**Inhibition experiment**. For opsin-mediated inhibition, Sim1-Cre mice containing eArchT3.0 in PVH^Sim1 neurons and optic fiber implants over LSv underwent two 2-hour trials: during the first trial, mice were measured food intake with food ad libitum conditions beginning at early morning. After at least one day recovery, the same feeding experiments was conduction with photo-illumination (556 nm, ~ 10 mW/mm²; constant-on for every other minute) during the testing period. Food intake was compared between the two sessions.

For DREADD-mediated inhibition, Vgat-Cre mice with delivery of AAV-fas-ChR2-eGFP to the PVH and of AAV-FLEX-Gi-DREADD-mCherry to the LSv were used for self-grooming and jumping behaviors elicited by photostimulation. The AAV-fas-ChR2-eGFP vectors express ChR2 selectively in non-Cre-expressing neurons and will therefore express in most PVH neurons as the majority of PVH neurons are glutamatergic[36]. Time spent in grooming and number in jumping elicited by photostimulation through optic fibers targeting LSv were compared between saline and CNO treatments. Post-hoc analyses were conducted to verify correct viral expression in the PVH and LSv as well as optic fiber implantation.

**RTPP**. For Real Time Place Preference Assays, a commutator (rotary joint; Doric, Québec, Canada) was attached to a patch cable via FC/PC adaptor. The patch cable was then attached to the optic fiber cannula ferrule end via a ceramic mating sleeve. Another patch cable containing FC/PC connections at both ends allowed the connection between the commutator and the 473 nm laser, which was controlled by the Master-8 pulse stimulator. Sim1-Cre, Sim1-Cre::Vglut2^flox/flox and controls, injected with Cre-dependent ChR2 or GFP viruses, respectively, and implanted with optic fibers above LSv, were placed in a clean 45 cm × 45 cm × 50 cm chamber equipped with a camera mounted on top of the chamber and optical fiber patch cable attached to a commutator. The testing chamber was wiped down with 70% isopropyl alcohol between tests. Prior to starting experiments, the patch cable was attached to optic fiber ferrule end of the mouse's cannula. At the start of the experiment, mice were placed in the light-off zone, in which no light was applied. Then, for twenty minutes, the mice were allowed to freely roam the enclosure, which was divided into two equal zones containing the light-off zone and a light-on zone, in which 5 Hz, 100 ms (473 nm, ~5 mW/mm²) light pulses were delivered. The side paired with photostimulation was counterbalanced between mice. For some experiments, instead of paring with one side, photostimulation was paired with the periphery. For other experiments, 12 hr fasted mice were used and photostimulation was paired with the side where chow food pellets were placed. Tracking data, including time spent in each zone, were collected and analyzed by Ethovision XT software (version 11.5; Noldus, Wageningen, Netherlands). Preference for side zones or center versus peripheral zones was determined by comparing the percentage of time spent in each zone.

**Resident-intruder assay**. Sim1-Cre male mice with AAV-FLEX-ChR2-eGFP delivery to the PVH and fiber optic implantation targeting LSv were used as residents. Resident mice were housed together with females before the experiment and resident mouse cages were not cleaned for a minimum of 1 day until the behavioral test to establish territory. On the test day, the tip of a fiber optic cable (200 μm in core diameter, ThorLabs) was inserted to a depth just above the target brain region by introducing the cable into the brain through the fiber optic. For the ferrule-connector system, a ferrule patch cord was coupled to a ferrule on the head using a zirconia split sleeve (Doric Lenses). The optical fiber was connected to a laser (473 nm; Opto Engine LLC, Midvale, Utah, USA) directly or via an optical commutator (Doric Lenses) to avoid twisting of the cable caused by the animal's movement. One to three male intruders were individually introduced to Sim1-Cre male residents in a testing session with a 3 min interval between intruders. Laser was switched on at 30 seconds after the introduction of an intruder and lasted 30 seconds. After laser was turn off, mouse behaviors were continuously recorded for addition 2 mins. Time spent in attack (chasing, mounting, and fighting) in the resident males was recorded for each intruder.

**Brain tissue preparation, imaging, and post hoc analysis**. After behavioral experiments were completed, study subjects were anesthetized with a ketamine/xylazine cocktail (100 mg/kg and 10 mg/kg, respectively) and subjected to transcardial perfusion. During perfusion, animals were flushed with 20 mL of saline prior to fixation with 20 mL of 10% buffered formalin. Freshly fixed brains were then extracted and placed in 10% buffered formalin in 4 °C overnight for post fixation. The next day, brains were transferred to 30% sucrose solution and allowed to rock at room temperature for 24 h prior to sectioning. Brains were frozen and sectioned into 30 μm slices with a sliding microtome and mounted onto slides for

post hoc visualization of injection sites and cannula placements. Mice with missed injections to the P or misplaced optic fibers over the LSv were excluded from behavioral analysis. Representative pictures of LSv and PVH injection sites and cannula placements were visualized with confocal microscopy (Leica TCS SP5; Leica Microsystems, Wetzlar, Germany).

**In vivo fiber photometry experiments**. *Sim1-Cre* mice with specific delivery of AAV-FLEX-GCaMP6m[62] to the PVH and optic fiber implantation targeting PVH neurons and *Vgat-Cre* mice with specific delivery of AAV-FLEX-GCaMP6m to the LSv and optic fiber implantation targeting LSv were used for the in vivo fiber photometry $Ca^{2+}$ imaging studies. In retrograde studies, AAVrg-EF1a-Cre-Venus viruses were injected to the LSv of wild-type mice and the AAV-Flex-GCaMP6m viruses were injected to the PVH, and the recording were performed 6 weeks after the surgery. The GCaMP6m virus was provided by the Baylor NeuroConnectivity Core. The experiments were conducted at least 4 weeks after the surgery. We used the fiber photometry system from Doric Lenses to monitor $Ca^{2+}$ signal from a group of PVH Sim1 neurons and LSv GABAergic neurons. Mice were acclimated to the behavioral chamber for at least 15 min prior to the beginning of each testing session. After baseline recording for 10 seconds, water spray was started by spraying one time with water toward the head of mice with a sprayer and the recording will continue for ~ 2 mins. In the resident-intruder assay, *Sim1-Cre*, and *Vgat-Cre* mice described above were used as intruder mice. Recording was started after the intruder was introduced in the resident cage, and mouse behaviors will also be simultaneously recorded. For feeding studies, fasted or well-fed mice were acclimated in the cage for at least 5 mins before one high-fat diet pellet was introduced to the cage. Mouse behaviors including feeding was videotaped simultaneously with photometry recording.

Data were acquired through Doric Studios V4.1.5.2. and saved as comma-separated files (header was deleted) at a sampling rate of either 1kS/s or 2.5kS/s. An in-house script (available upon request) written in R (V3.4.4. "Someone to Lean On", packages ggplot2, reshape, zoo, plyr, viridis, scales) was used to calculate the baseline fluorescence (F0) using linear regression (c.f. (36)) across a chosen period of recording. The change in fluorescence (dF) was then determined from the residuals and multiplied by 100 to arrive at % dF/F0. For water spray, loud sound, and light, the baseline was calculated from 10 seconds prior to stimulus onset, to 20 seconds after stimulus onset. For feeding, because mice typically engage in a series of quick successive feeding bouts, the F0 was calculated across a 50-second period with the feeding bout in the center. The average % dF/F0 across the whole period was compared with the average % dF/F0 during the feeding bout. The calcium-independent fluorescence signal was also recorded; no significant alteration in signal level was observed compared with calcium-dependent signal (indicating steady light-emitting diode excitation), as shown in gray lines of Fig. 9i, j.

**Statistics**. GraphPad Prism 7.00 (GraphPad Software, Inc., La Jolla, CA, USA) was used for all statistical analyses and construction of graphs. Two-way repeated measures or regular analysis of variance (ANOVA) followed by Dunnett's or Sidak's multiple comparisons test were used for group comparisons. Single-variable comparisons were made by paired or unpaired two-tailed Student's *t* tests, or one ANOVA followed by Tukey's multiple comparison post hoc tests. Error bars in graphs were represented as ± SEM.

**Reporting summary**. Further information on research design is available in the Nature Research Reporting Summary linked to this article.

## Data availability
The authors declare that the data supporting the findings of this study are available within the paper and supplementary figures. The source data associated with Fig. 5b, e are provided as a Source Data file.

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

## Acknowledgements

This study was supported by NIH R01 DK114279 and NIH R21NS108091 to Q.T.; NIH 5F31DA041703 to L.R.M.; NIH R01DK117281 and R01DK101379 to Yong. X.; and R01DK109934 to B.R.A. and Q.T. We also acknowledge the Neuroconnectivity Core funded by NIH IDDRC grant 1 U54 HD083092 for providing AAV vectors. We acknowledge the Tong lab members for helpful discussion, Dr. Zhengmei Mao for help with microscopy. Q.T. is the holder of Cullen Chair in Molecular Medicine at McGovern Medical School.

## Author contributions

Y.X., Y.L. and R.M.C. conducted the research with help from L.R.M., C.Z., X.H. and J.Z.; Y.X. and R.M.C. analyzed the data; B.R.A., N.J.J. and Yong. X. provided essential reagents; Q.T. and Y.X. conceived and designed the studies, and wrote the manuscript with significant inputs from N.J.J., Yong. X. and B.R.A.

## Additional information

**Competing interests:** The authors declare no competing interests.

