## [Peer Review File · Nature Communications]

Reviewers' Comments:

Reviewer #1:

Remarks to the Author:

In Xu et al's paper, they used a series of gain and loss of function tools to dissect the circuits between PVH to septum in modulating feeding and emotional responses. They demonstrated that the PVH neurons project to LSv neurons, and that weak stimulation of PVH terminals in LSv elicits stress-related self-grooming while strong stimulation induces fear escape jumping with feeding suppression. In particular, inhibition of glutamatergic inputs to LSv increases feeding with reducing anxiety. The authors propose that the PVH to LSv circuit underlies comorbidity between eating and psychiatric disorders. It is clear that there is a huge amount of works went into the paper. While these results are interesting, I have a number of major conceptual and methodological concerns regarding this work that significantly reduce my enthusiasm for it.

1. The most important demonstration that is necessary for claiming that LSv are involved in the behaviors. The LS is a large structure, containing over 10 different compartments. Even roughly speaking, it contains rostral, caudal, dorsal and ventral subdivisions. Each compartment receive different projection patterns and probably functions. In this study, it appears that different LS subdivisions are involved in different experiments but then the results were considered together to draw the conclusion. For example, the infections in the different figures cover all the whole LS, even the whole septum. So, how could one draw the conclusion that LSv mediated the behaviors by stimulating the PVH fibers? For example, in figures 2 and 5, PVH fibers are also apparently observed in the MS, and in the septofimbrial nucleus and median preoptic nucleus as well. All these issues raise questions about the structures under investigation in this study. In general, the reviewer suggests the authors pay more attention in targeting the structure and quantifying the histology to make sure that the observed effect is due to the region under investigation with photostimulation.
2. Based on the few examples of histology in the paper, there appears to be significant labeling in the medial septum and other subdivisions. Do the authors want to make the strong claim that the effects are specific to lateral septum (LSv), as is stated in the manuscript? If so, it would be good to quantify the extent of neurons with c-fos with PS specifically within the LSv and show a correlation between the extent of expression and the behavioral effect. Similarly, it would also be good to show no correlation between medial septum expression and behavior. Such a demonstration would add a lot of rigor to the main claims of the manuscript. In its absence, it is hard to be certain that the observed effects are specifically due to LSv.
3. The authors claim that local infusion of AP5/DNQX in LSv displayed reduced anxiety and suggest that blocking ongoing glutamatergic action in LSv neurons reduces anxiety (Fig. 4i and Fig S5), which is over-stated because how the authors confirmed the infused AP5/DNQX is limited inside the LSv without spreading to adjacent brain regions?
4. The mechanisms of different behaviors resulting from weak and strong stimulations should be substantially examined and explained, which is pretty confusing. If so, what is the exact interface? and how to explain it?
5. Many control experiments are missing in the Figures. Particularly, lack of a controls makes it hard to interpret the specificity of food intake suppression and anxiety behaviors. If a manipulation induces nausea or a headache might that not suppress food intake yet have no direct effect on food intake?

Reviewer #2:

Remarks to the Author:

In this interesting study, Tong and colleagues have identified a glutamatergic projection from the paraventricular nucleus of the hypothalamus (PVH) to ventral lateral septum (LSv) that potently suppresses feeding and enhances defensive behaviors. Apart from the general significance of defining behaviorally relevant circuits, this work is potentially important as the projection identified here may provide a new entry point into understanding how appropriate behavioral choices are made in the face of competing motivational drives (e.g. hunger and defense), and how dysfunction in such circuits may lead to neuropsychiatric disorders.

Issues

1. The major issue which is not addressed either experimentally or in the discussion is the question of whether there is a single population of neurons that, in an intensity-dependent manner, promotes both inhibition of feeding and defensive behaviors, or if alternatively there are distinct subsets of PVH>LSv neurons which regulate these distinct behavioral functions. In fact, their data provide support for the latter model, as stimulation of CRHR1(PVH>LSv) projections was found to be aversive but did not affect feeding (Supp Fig 9). Complete resolution of this issue (i.e. molecular identification of the specific PVH subsets that mediate specific behavioral effects observed when the entire PVH>LSv projection is manipulated) is complex and should be the focus of follow-up studies. However, as this study represents the first description of the PVH>LS projection, a more thorough characterization of the neurons comprising this projection would be appropriate here, and ideally would include determination of: a) percentage of PVH cells that project to LSv, where exactly in the PVH these LS projectors are located, and whether or not they map on to one of the specific PVH subdomains defined by Hongwei Dong's group (Biag et al, 2010 JCN); b) the precise location of the PVH projections in the LS (i.e. which AP levels of LSv contain PVH Sim1+ axons, and what are the relative densities of these axons at the different AP levels); and c) determination of whether PVH>LSv neurons project only to LSv, or if they also send collaterals to other regions. The issue of whether the same or distinct PVH subsets mediate the different effects observed by optogenetic manipulations should also be addressed in the discussion. Lastly, a direct test of whether the same or distinct neuronal subsets are engaged by stimuli that only interrupt feeding vs. those that elicit robust defensive behaviors (e.g. jumping) would be helpful for dissociating the two alternative models (e.g. using Fos catFISH analysis or single cell GCaMP imaging of PVH>LSv projectors).

2. The imaging data presented are suggestive but are not specific for the PVH>LSv projection. That is, in hypothalamus they are imaging from the entire PVH population (only a fraction of which projects to LSv), and in LS, they are imaging from all GABAergic neurons (only a fraction of which receive input from PVH). In order for this data to be interpretable, the authors should either image from PVH axon terminals in LSv, or they should image from PVH neurons retrogradely labeled based on their projection to the LS (e.g. using canine adenovirus or AAVretro). In addition, the imaging data are uninterpretable in the absence of statistical analysis (e.g. permutation testing) and negative controls (e.g. mice expressing GFP instead of GCaMP, to rule out motion artifacts).

3. For the grooming assays, it would be helpful to explain how exactly 'incorrect transitions' and 'interrupted bouts' were scored so that other groups attempting to replicate these procedures have precise definitions and can apply the same criteria.

Reviewer #3:

Remarks to the Author:

This is very interesting work in a relatively untapped field, with well controlled studies showing the novel finding that specifically glutamatergic input from PVH—>LSV bidirectionally affects feeding and anxiety like behavior. The authors should be commended for succinctly summarizing their vast amount of work spanning pharmacology, circuitry, and behavior studies, good use of appropriate controls for most of the studies, and the novelty of the finding. The videos are helpful and the behavior change in response to stimulation is striking. The focus on glutamate is particularly interesting, and timely in relation to the psychiatric neuroscience shift of focus from neuropeptide to neurotransmitter influence on disease, and would benefit from further discussion, including how to make sense of the model that includes combined gaba-ergic input coming from the PVH. My enthusiasm for publication of this manuscript in its current form is somewhat tempered by the over-reaching conclusions and generalizations in the description and interpretation of the results, which are not well supported by the data.

My major concerns with the manuscript include:

Overall, the wording of the manuscript is rather inconsistent and overly anthropomorphic. It is nearly impossible to interpret animal behavior as “negative emotions” or “anxiety” outright. I recommend the use of anxiety-like behavior, response to negative valence, etc in these settings, or just describing the behavior as it is (i.e. exploratory, light-averse etc).

Along the same lines, there is some mis-interpretation of certain assays and inconsistency in the use of the wide variety of assays used. For example, there seems to be some confusion of the use of fear and anxiety terms in describing animal behavior; based on the introduction it seems the authors would like to study a fear circuit, but based on the assays used it seems the authors are studying anxiety-like behavior, or differences in exploratory activity, locomotor activity, aggression, and valence-based preference. Minimal evidence is given for the use of the “shelter hut” assay to study “fear” behavior, and in fact, it is no different than the light-dark or center-perimeter studies for anxiety-like/exploratory and locomotor behavior. In general, if one is using multiple assays to strengthen the argument, it is best practice to use the same assays for each of the conditions to allow for best comparisons and conclusions to be drawn. The authors use different assays for different experiments (for example, they only use the resident-intruder assay in the original photostim studies, but do not report its use with the pharmacologic or occlusion studies that follow). Similarly, the use of multiple different time points for fasting should be better justified – the 6h to 24h difference in scalability is interesting, but no justification is given for the 12h fast used in later experiments. Both of these sets of data are used to make overly strong conclusions, which would be better supported by consistent use of behavior experiments and timing. Furthermore, this response to different stimulation pulse lengths would also benefit from further discussion, and perhaps some thought as to the mechanism, or how the mechanism would be of use for translation.

Also in regard to interpretation of different assays, it seems that on the assays that allow the mice to move freely around a cage (center-perimeter / light-dark / shelter hut), there is a difference in the distance traveled, which calls into question the nature of the behavior being studied (if locomotor activity is changed, it cannot be interpreted as “anxiety-like behavior”). Notably, the 100ms stim leads to further distance traveled than does the 10ms stim, so that is similarly scalable, but may have nothing to do with anxiety-like behavior or fear. It is unclear how the authors chose the “environmental stressors” to integrate for their photometry studies, and they may or may not be relevant to the stated goal. The assay used to study competition between hunger and negative valence is confounded because the authors already showed that stimulation of PVHLSV decreases feeding, so that would obviously result in decreased feeding regardless of valence and RTPP. A better experiment to study overcoming an innate fear response would be to use predator odor, or shock, and furthermore, an important experiment to show that the circuitry in question does in fact allow for

overcoming fear and inducing feeding would be to inhibit the PVHLSV circuit in the setting of an actual threat (odor or shock) with food available to the food restricted mouse, as done in Burnett et al Neuron 2016, or Jikomes et al Curr Biol 2016 where the interaction between fear-related behavior and feeding was similarly investigated. Finally, the authors overstate the data from their photometry experiments as well – these data cannot really be used to comment on dynamics/kinetics of circuit direction or activity. Photometry averages across the whole population and has slower rise times than unit recording, so it is not adequate to measure latency between brain areas, nor is it appropriate to infer one population drives the other.

Another related concern is the confusion of arguments in human translation. The theory on antipsychotic meds causing weight gain is not supported by the arguments presented in the introduction and the discussion of this manuscript. There may be an interesting new take on the circuitry involved, but it is not well backed up by the data here. I recommend leaving those statements out of the intro and discussion so as not to dilute the strength of the findings as they are. A common neural pathway and reciprocal control are not the same thing (last 2 lines of the intro). The discussion of eating disorder pathology is also opposite generally accepted clinical theory, though this too is a topic of active discussion; the current manuscript seems to state that the change in eating precedes anxiety in anorexia (line 5 of the intro), though in general it's thought that the decrease in food intake is meant to modulate some perceived stress (whether real, fear-related, or anxiety-based). The abstract similarly makes a claim that in anorexia, "anxiety dominates over hunger" and that is an oversimplified view of the clinical complexity. The intro similarly oversimplifies the findings of the feeding/emotion studies based in the amygdala and lateral hypothalamus, which leads to potential misinterpretation and does not do justice to that body of work, nor does it come across as relevant to the results of this paper (there is no mention of how the PVHLSV circuitry may relate to the known circuitry, i.e. LSVLH or amygdala). The introduction and conclusion would benefit from significant re-writing.

Throughout the description of the studies, the authors do not make their methodology clear. For example, for the studies of figure 2, was the optical stimulation unilateral or bilateral? For figure 3, which mice were used? Were both short and long pulses used? For figure 4 it would be useful if the manuscript overtly stated which part of the field was paired to the stimulation. For supplementary figures 6-8 in the results description it would be best to state what was seen in the experiment: the current statement "failed to find evidence for a contribution... through oxytocin, CRH, AVP neurons" is confusing to interpret; it would be more precise and descriptive to say "failed to find anterograde projections" from those neurons. For figure 7, what do different colors mean (different mice? Different days?). In the methods section it says that both male and female mice were used. Given that the PVH is a highly sexually dimorphic region, it would be important to distinguish for what experiments each sex of mouse was used.

Finally, I do think the authors may be overstating the novelty of the circuit based investigation. There has been some work on projections from the hypothalamus to the forebrain, though not yet characterized using modern tools (see Sawchenko/Swanson papers from the 80s, and Simerly papers from the 2000s), Furthermore there has been some work done on the reciprocal connections between LSVhypothalamus (Sheehan et al 2004, Sweeney et al 2016). This doesn't dampen the novelty of their work, but I recommend that they temper their description of it to reflect prior work.

Reviewer #4:

Remarks to the Author:

Manuscript Number: NCOMMS-18-19411

In this manuscript, Xu and co-workers investigated a novel neurocircuit underlying the interplay between stress-related and feeding behaviours. They nicely mapped and characterized a novel neurocircuit originating from glutamatergic neurons located in the PVH that project to GABAergic neurons in the LS to control fear and appetite.

This manuscript provides interesting and novel data regarding the neurocircuits controlling feeding depending on the emotional responses. This findings will generate a strong interest in the field of research.

Overall, the experiments appear well executed (including statistical analysis) and conclusions drawn by the authors are supported by experiments provided in the paper. Some aspects of the manuscript could be improved:

1. As the authors always compared weak (10ms) and strong stimulation (100ms), for consistency, both should be included for all experiments throughout the manuscript. For instance, in Figure 5 the feeding is not depicted for the 100ms stimulation after 6hours fasting. This aspect will also be interesting for the supplementary figures 6 and 7 to rule out that distinct populations could mediate the behaviour observed with the 10ms stimulation vs the 100 ms.
2. In figure 3d-e, the author should also provide the behaviour experiments with optogenetics + CNO only as well as optogenetics + AAV-Fas-ChR2 only.
3. Figure 5 a-b: would stimulation also change the steady food intake (i.e in random fed animals as in 5f/h) ?
4. In general, details are missing in the main text to improve the readability. For instance, when the author describes the figure 3d-f, the mouse line used is not included in the main text but only in the legends of the supplementary figures.
5. In supplementary figure 1, the annotation "wild-type mice n=14" is confusing/misleading.
6. In supplementary figure 3, a "light-OFF" control should be included to compare the c-Fos induction to basal. This aspect is important to further understand the differences observed with 10 and 100ms stimulation.
7. The c-Fos immunostaining depicted in Figure S3 and S4 cannot be visualized on the representative pictures.
8. More discussion on the potential PVH neuronal phenotype involved neurons could be included.

Responses to reviewers' comments

We would like to thank all reviewers for their insightful comments, to which we have performed substantial additional work to address. In short, we have conducted detailed retro- and antero-grade tracing experiments to document specific PVH to LSv projections, and importantly we have reproduced the behavioral results and *in vivo* neuron activity changes from selective LSv-projecting PVH neurons. In addition, we have also conducted other necessary experiments and analysis to address other comments from the reviewers. Accordingly, we have modified some of the original main figures and added 2 new main figures (Figs. 8 and 9). With this additional data and revision, we think that our manuscript has been significantly improved and the reviewers' comments have been sufficiently addressed. The following are our point-to-point responses to the reviewers' comments.

Comments from Reviewer 1:

1. The most important demonstration that is necessary for claiming that LSv are involved in the behaviors. The LS is a large structure, containing over 10 different compartments. Even roughly speaking, it contains rostral, caudal, dorsal and ventral subdivisions. Each compartment receive different projection patterns and probably functions. In this study, it appears that different LS subdivisions are involved in different experiments but then the results were considered together to draw the conclusion. For example, the infections in the different figures cover all the whole LS, even the whole septum. So, how could one draw the conclusion that LSv mediated the behaviors by stimulating the PVH fibers? For example, in figures 2 and 5, PVH fibers are also apparently observed in the MS, and in the septofimbrial nucleus and median preoptic nucleus as well. All these issues raise questions about the structures under investigation in this study. In general, the reviewer suggests the authors pay more attention in targeting the structure and quantifying the histology to make sure that the observed effect is due to the region under investigation with photostimulation.

Response: The reviewer raised a concern on potential other lateral septum structures other than the ventral part in mediating the behavior described in this manuscript. To examine this, we have conducted an unbiased projection survey in the LS. We delivered AAV-FLEX-Syn-GFP vectors to the PVH of Sim1-Cre mice and examined all sections that contained GFP-positive fibers in the lateral septum. As illustrated in the new supplementary Fig. 1, with the AAV vectors delivered to all PVH neurons, GFP-positive fibers were found only in the ventral part of LS (LSv), and as a matter of fact, LSv is the only brain structure that receives projections from the PVH, as evidenced from a series of sections from rostral to caudal dimension with GFP-positive fibers. In addition, as shown in the new Fig.8, we delivered a large amount of Cre-dependent AAV-DIO-mCherry to the LS of mice that received the anterograde tracer AAV1-Cre-Venus viral vector delivery to the PVH, and only LSv neurons exhibited the mCherry expression. These new results demonstrated that the PVH selectively projects to LSv.

2. Based on the few examples of histology in the paper, there appears to be significant labeling in the medial septum and other subdivisions. Do the authors want to make the strong claim that the effects are specific to lateral septum (LSv), as is stated in the manuscript? If so, it would be good to quantify the extent of neurons with c-fos with PS specifically within the LSv and show a correlation between the extent of expression and the behavioral effect. Similarly, it would also be good to show no correlation between

medial septum expression and behavior. Such a demonstration would add a lot of rigor to the main claims of the manuscript. In its absence, it is hard to be certain that the observed effects are specifically due to LSv.

Response: This concern is the same as above and has been addressed by our additional data in the new Supplementary Fig. 1. The reviewer suggested to use c-Fos expression to document immediate downstream neuron activation. As a matter of fact, we have done the experiment (Supplementary Fig. 3), where we observed an increase in c-Fos expression by increasing the strength of photostimulating PVH to LSv fibers. However, relying on c-Fos expression to survey monosynaptic downstream neurons can sometime be misleading because 1) c-Fos is a general marker of any neuron that becomes active around the time during tissue harvesting and as shown in our photometry data, LS neurons are very sensitive to stressors, and a lot of LS neurons may become activated regardless of photostimulating PVH to LSv fibers; and 2) as demonstrated in Fig. 1, LS neurons are known to form recurrent inhibitory projections within themselves, and it is conceivable that changes in the activity of one group of LS neurons (e.g. LSv neurons) might change the activity of nearby neurons, leading to alterations of c-Fos expression.

3. The authors claim that local infusion of AP5/DNQX in LSv displayed reduced anxiety and suggest that blocking ongoing glutamatergic action in LSv neurons reduces anxiety (Fig. 4i and Fig S5), which is over-stated because how the authors confirmed the infused AP5/DNQX is limited inside the LSv without spreading to adjacent brain regions?

Responses: The reviewer has raised a valid concern on potential AP5/DNQX infusion outside the LSv region. However, our goal here is to test whether glutamate receptors mediate the behavior, and our results support that activation of glutamate receptors in the PVH projection area is required for the observed behavior.

4. The mechanisms of different behaviors resulting from weak and strong stimulations should be substantially examined and explained, which is pretty confusing. If so, what is the exact interface? and how to explain it?

Responses: The reviewer raised an interesting question that we are also interesting to understand. A similar phenomenon has also been reported previously (1, 2). Our current hypothesis is that strong stimulation may result in great recruitment of downstream neurons, as supported by more c-Fos expression in the LSv by higher intensity stimulation (Supplementary Fig. S3), and thereby reaching a threshold in changes of neuron activity in the downstream pathway that trigger different behaviors. We agree with the reviewers that more substantial experiments are certainly required to understand the underlying biology. However, these experiments, given the scope of work presented here, are beyond the scope of this manuscript. In addressing this concern, we have added a paragraph in Discussion focusing on this point.

5. Many control experiments are missing in the Figures. Particularly, lack of a controls makes it hard to interpret the specificity of food intake suppression and anxiety behaviors. If a manipulation induces nausea or a headache might that not suppress food intake yet have no direct effect on food intake?

Response: We think we have included all necessary controls in our experiments. We have included GFP control or KO mice for ChR2 effects, pre-light and post-light for light effects in most experiments. In our studies on food intake suppression, we even included a “dose”

dependent studies, with mice fasted over different hours, and observed a “dose” dependent effect. Thus, our conclusion is robust. The reviewer raised a concern on potential “nausea” or “headache”. Although we couldn’t rule out completely “nausea” or “headache” as both are difficult to measure, we think that is unlikely because the negative valence and food intake reduction induced by photostimulation of PVH to LSv fibers were both rapid and reversible (Figs. 5b and 5e). Also the transition from non-attach to attach by switching off light stimulation occurred within a few seconds (Fig. 2i). The rapid reversibility within a few seconds argues against a “nausea” or “headache”, which normally takes a relatively long time to recover. On the other hand, even if mice are experiencing “nausea” or “headache”, these might be phenotypic manifestations of negative valence elicited by photostimulating PVH to LSv fibers, which would still be a significant finding.

Comments from Reviewer 2:

In this interesting study, Tong and colleagues have identified a glutamatergic projection from the paraventricular nucleus of the hypothalamus (PVH) to ventral lateral septum (LSv) that potently suppresses feeding and enhances defensive behaviors. Apart from the general significance of defining behaviorally relevant circuits, this work is potentially important as the projection identified here may provide a new entry point into understanding how appropriate behavioral choices are made in the face of competing motivational drives (e.g. hunger and defense), and how dysfunction in such circuits may lead to neuropsychiatric disorders.

Response: We would like to thank the reviewer in appreciating the significance of our work.

1. The major issue which is not addressed either experimentally or in the discussion is the question of whether there is a single population of neurons that, in an intensity-dependent manner, promotes both inhibition of feeding and defensive behaviors, or if alternatively there are distinct subsets of PVH>LSv neurons which regulate these distinct behavioral functions. In fact, their data provide support for the latter model, as stimulation of CRHR1(PVH>LSv) projections was found to be aversive but did not affect feeding (Supp Fig 9). Complete resolution of this issue (i.e. molecular identification of the specific PVH subsets that mediate specific behavioral effects observed when the entire PVH>LSv projection is manipulated) is complex and should be the focus of follow-up studies. However, as this study represents the first description of the PVH>LS projection, a more thorough characterization of the neurons comprising this projection would be appropriate here, and ideally would include determination of: a) percentage of PVH cells that project to LSv, where exactly in the PVH these LS projectors are located, and whether or not they map on to one of the specific PVH subdomains defined by Hongwei Dong’s group (Biag et al, 2010 JCN); b) the precise location of the PVH projections in the LS (i.e. which AP levels of LSv contain PVH Sim1+ axons, and what are the relative densities of these axons at the different AP levels); and c) determination of whether PVH>LSv neurons project only to LSv, or if they also send collaterals to other regions. The issue of whether the same or distinct PVH subsets mediate the different effects observed by optogenetic manipulations should also be addressed in the discussion. Lastly, a direct test of whether the same or distinct neuronal subsets are engaged by stimuli that only interrupt feeding vs. those that elicit robust defensive behaviors (e.g.

jumping) would be helpful for dissociating the two alternative models (e.g. using Fos catFISH analysis or single cell GCaMP imaging of PVH>LSv projectors).

Response: We thank the reviewer for raising these important and interesting further studies to delineate the underlying projection between PVH and LSv. As shown in new Fig. 8, we have used retro-AAV vectors to trace PVH projecting neurons, which are mainly located in the anterior part of PVH, and a less number of neurons in the medial part of the PVH and almost no neurons found in the posterior part of the PVH. We have also performed anterior AAV vector tracing with AAV1-Cre-Venus, and presented the result in the new Fig. 8, which showed postsynaptic neurons in the LSv. We have labelled bregma levels in these tracing studies, which have significantly improved delineation of the PVH to LSv projection. Given the inherent variability in these tracing studies caused by slight variation in AAV vector injections and in time duration required for vector tracing, conclusion from detailed counting of labelled neurons in the PVH or LSv might be misleading. We think that this will not jeopardize our current conclusion in this study.

The reviewer raised an interesting possibility that distinct population of PVH neurons that send projections to LSv to mediate different behaviors (feeding inhibition and defensive behaviors) observed in this study. We think this is very unlikely because 1) in all animals tested in this study, they exhibited the changes in feeding and behavior simultaneously and we failed to identify one animal that only showed one effect but not the other; 2) our new data from stimulation of LS fibers from selective LSV-projecting PVH neurons also elicited feeding inhibition and defensive behaviors simultaneously; and 3) the effects on feeding inhibition and behaviors are “dose” dependent, as shown by the data shown in Figs, 2d and 2f, Fig. 4e and Fig. 5a, where the extent of hunger by various fasting hours changed the feeding inhibition by photostimulation and different levels of photostimulation caused different behaviors (self-grooming and jumping) with different degrees of aversion. More studies are certainly required for further dissect the projection. The data presented in Supplementary Fig. 9 might be misleading because 1) the effect on self-grooming and aversion is small compared to the one presented in Figs. 2 and 4 (note that the long duration 100ms stimulation used here produced frantic jumping in Sim1-Cre mice); 2) CRHR1-Cre targeted some neurons outside the PVH (Supplementary Fig. S9a); 3) there is a trend for feeding inhibition (Supplementary Fig. S9d). Given the high level of sensitivity of PVH neurons with c-Fos expression in response to stress, it would be extremely difficult to rely on c-Fos with the suggested CatFish method. The single-neuron Ca²⁺ imaging experiment will offer insights and it appears to be an interesting follow up study. Per the reviewer’s suggestion, we have added this interesting point to Discussion.

The Hongwei Dong’s paper categorized the PVH neuron groups using retrograde tracing mainly from ascending projections/pathways whereas ours on efferent ascending pathways. Given our new data on rostral PVH as a main location for the origin of projection, it is difficult to align the two pieces of information together. However, we have added one sentence in Discussion regarding the relation on LSv-projecting and hindbrain- and spinal cord-projecting PVH neurons.

2. The imaging data presented are suggestive but are not specific for the PVH>LSv projection. That is, in hypothalamus they are imaging from the entire PVH population (only a fraction of which projects to LSv), and in LS, they are imaging from all GABAergic neurons (only a fraction of which receive input from PVH). In order for this data to be

interpretable, the authors should either image from PVH axon terminals in LSv, or they should image from PVH neurons retrogradely labeled based on their projection to the LS (e.g. using canine adenovirus or AAVretro). In addition, the imaging data are uninterpretable in the absence of statistical analysis (e.g. permutation testing) and negative controls (e.g. mice expressing GFP instead of GCaMP, to rule out motion artifacts).

Response: We thank the reviewer for this helpful suggestion. As shown in the new Fig. 9, we have performed the experiment and demonstrated that the activity of LSv-projecting PVH neurons exhibit a similar response to stressors and during feeding. This data has improved our conclusion significantly. We have also revised our data presentation to include statistical analysis, as shown in the new Fig. 7i.

3. For the grooming assays, it would be helpful to explain how exactly ‘incorrect transitions’ and ‘interrupted bouts’ were scored so that other groups attempting to replicate these procedures have precise definitions and can apply the same criteria.

Response: We have clarified this issue and cited papers from others and ourselves.

Comments from Reviewer 3:

This is very interesting work in a relatively untapped field, with well controlled studies showing the novel finding that specifically glutamatergic input from PVH—>LSV bidirectionally affects feeding and anxiety like behavior. The authors should be commended for succinctly summarizing their vast amount of work spanning pharmacology, circuitry, and behavior studies, good use of appropriate controls for most of the studies, and the novelty of the finding. The videos are helpful and the behavior change in response to stimulation is striking. The focus on glutamate is particularly interesting, and timely in relation to the psychiatric neuroscience shift of focus from neuropeptide to neurotransmitter influence on disease, and would benefit from further discussion, including how to make sense of the model that includes combined gaba-ergic input coming from the PVH.

Response: We thank the review in appreciating the significance of our work.

1) Overall, the wording of the manuscript is rather inconsistent and overly anthropomorphic. It is nearly impossible to interpret animal behavior as “negative emotions” or “anxiety” outright. I recommend the use of anxiety-like behavior, response to negative valence, etc in these settings, or just describing the behavior as it is (i.e. exploratory, light-averse etc).

Response: We appreciate the reviewer the concern on using the description on human emotion to describe mouse behaviors related to animal valence. This is certainly an area of debate, as nicely illustrated a recent review by Tye KM et al, Neuron, 2018(3). Emerging studies suggest that the underlying neurocircuits controlling behaviors related to negative valence are conserved between rodents and humans (3). In addition, some have proposed that human emotions may also be applicable to rodents (4). There are some published papers in the field describing

mouse using “anxiety”. For example, studies from Dr. David Anderson (5) and from Dr. Kay Tye (6). Emotional states have also been used in rodents (7). We agree with the reviewer and to rule out potential confusion on this debatable issue to some readers, we have cited the Tye KM paper and changed wording to avoid direct description of anxiety or emotion on mice throughout the manuscript.

2) Along the same lines, there is some mis-interpretation of certain assays and inconsistency in the use of the wide variety of assays used. For example, there seems to be some confusion of the use of fear and anxiety terms in describing animal behavior; based on the introduction it seems the authors would like to study a fear circuit, but based on the assays used it seems the authors are studying anxiety-like behavior, or differences in exploratory activity, locomotor activity, aggression, and valence-based preference. Minimal evidence is given for the use of the “shelter hut” assay to study “fear” behavior, and in fact, it is no different than the light-dark or center-perimeter studies for anxiety-like/exploratory and locomotor behavior. In general, if one is using multiple assays to strengthen the argument, it is best practice to use the same assays for each of the conditions to allow for best comparisons and conclusions to be drawn. The authors use different assays for different experiments (for example, they only use the resident-intruder assay in the original photostim studies, but do not report its use with the pharmacologic or occlusion studies that follow). Similarly, the use of multiple different time points for fasting should be better justified – the 6h to 24h difference in scalability is interesting, but no justification is given for the 12h fast used in later experiments. Both of these sets of data are used to make overly strong conclusions, which would be better supported by consistent use of behavior experiments and timing. Furthermore, this response to different stimulation pulse lengths would also benefit from further discussion, and perhaps some thought as to the mechanism, or how the mechanism would be of use for translation.

Response: In this study, we argue a general role of the PVH projection to LSv in promoting a state of negative valence and its associated behavioral signs, including anxiety, fear and defense. Our results from the measurements of various behaviors support that activation of this projection promotes negative valence and inhibits feeding in a scalable fashion. Our goal here is not to specifically focus on anxiety, fear or defense. However, anxiety, fear and defense are inherently related as anxiety is a generalized state of fear. Our data support that these different states can be driven by the same neural pathway.

The shelter assay was adopted from a previous study from Dayu Lin’s group (8). In the original set up, the shelter was placed in the corner of the cage, which in our hands, most mice made the trips to the shelter directly, resulting in a small window for us to observe potential effects. We changed it to the center of the cage, and in this case, mice have to make efforts to enter to the shelter by overcoming anxiety associated with the center approach, thus measuring the mouse behavior in actively searching for safety and thus better indicating a state of fear.

We agree with the reviewer that different assays should be used to support one common argument. This is why we designed different assays (dark-light, center/corner, different fasting time, etc) to support that the PVH to LSv projection produces negative valence and feeding inhibition. For the same purpose, we also used the intruder assay to document the opposite that activation of the PVH to LSv projection reduces aggression. However, it is almost impossible for us to use the resident intruder paradigm in our pharmacological studies. The optogenetics

approach nicely activates the fibers acutely and reversibly without physically stressing mice, which is required to observe the acute changes in aggression behavior within the same animal before and after light stimulation, allowing a robust conclusion to be drawn. The pharmacology will have to compare animals in different experiments, which, given the inherent variation in animal aggression, is difficult to achieve a reliable comparison.

3) Also in regard to interpretation of different assays, it seems that on the assays that allow the mice to move freely around a cage (center-perimeter / light-dark / shelter hut), there is a difference in the distance traveled, which calls into question the nature of the behavior being studied (if locomotor activity is changed, it is cannot be interpreted as “anxiety-like behavior”). Notably, the 100ms stim leads to further distance traveled than does the 10ms stim, so that is similarly scalable, but may have nothing to do with anxiety-like behavior or fear. It is unclear how the authors chose the “environmental stressors” to integrate for their photometry studies, and they may or may not be relevant to the stated goal. The assay used to study competition between hunger and negative valence is confounded because the authors already showed that stimulation of PVHLSV decreases feeding, so that would obviously result in decreased feeding regardless of valence and RTPP. A better experiment to study overcoming an innate fear response would be to use predator odor, or shock, and furthermore, an important experiment to show that the circuitry in question does in fact allow for overcoming fear and inducing feeding would be to inhibit the PVHLSV circuit in the setting of an actual threat (odor or shock) with food available to the food restricted mouse, as done in Burnett et al Neuron 2016, or Jikomes et al Curr Biol 2016 where the interaction between fear-related behavior and feeding was similarly investigated. Finally, the authors overstate the data from their photometry experiments as well – these data cannot really be used to comment on dynamics/kinetics of circuit direction or activity. Photometry averages across the whole population and has slower rise times than unit recording, so it is not adequate to measure latency between brain areas, nor is it appropriate to infer one population drives the other.

Response: First we didn't use dark-light tests in this study. **We have to point out that our purpose of using center/perimeter and shelter assay is not to test if photostimulation increases anxiety.** If this is the case, then the opposite conclusion will be reached since photostimulation actually increased time spent in the center/shelter. In the center/perimeter assay, we paired photostimulation with peripheral stay and the animals spent much more time in the center by overcoming anxiety-associated with the center-stay, arguing that photostimulation caused a strong negative valence that is able to overcome the anxiety associated with center stay. The increases in locomotion may reflect the negative valence and animals actively looking for ways to avoid the perceived negative valence. The same is true for the shelter assay, in which the animals spent more time in the shelter placed in the center by overcoming the anxiety associated with the center stay.

Our data suggested that both PVH and LSv neurons rapidly responded to water-spray, loud sound and light, representing common changes in the environment, suggesting that these neurons are sensitive to stressors in the environment. These results support that stressors such as loud sound and light, normally eliciting fear/danger (similar to light-dark room tests), may induce stress-response/fear/anxiety by activating PVH and LSv neurons.

We want to point out that there is a fundamental difference between tests for competition of hunger and negative valence and tests for photostimulation in feeding inhibition. The former involves a choice between feeding with photostimulation and hunger with no stimulation, and the latter involves no choice. The reviewer suggested an interesting experiment involving natural fear. However, this paradigm doesn't fit our study since in our case photostimulation of PVH to LSv projections increases anxiety/fear, not the opposite. The experimental paradigm with a setting of natural fear can be used for an experiment to test whether activation increases behaviors related to aggression or reduces behaviors related to anxiety/fear, as conducted in the studies listed by the reviewer.

Our photometry data suggest that both PVH and LSv neurons are sensitive to stressors. This data, in combination with our other data suggesting that PVH neurons send glutamate projections to LSv, support that PVH and LSv are involved in the same pathway mediating responses to stressors and in turn controlling feeding. I agree with the reviewer that fiber photometry recording data alone is not sufficient to draw the conclusion. Accordingly, we have removed the sentence in Result.

4) Another related concern is the confusion of arguments in human translation. The theory on antipsychotic meds causing weight gain is not supported by the arguments presented in the introduction and the discussion of this manuscript. The may be an interesting new take on the circuitry involved, but it is not well backed up by the data here. I recommend leaving those statements out of the intro and discussion so as not to dilute the strength of the findings as they are. A common neural pathway and reciprocal control are not the same thing (last 2 lines of the intro). The discussion of eating disorder pathology is also opposite generally accepted clinical theory, though this too is a topic of active discussion; the current manuscript seems to state that the change in eating preceeds anxiety in anorexia (line 5 of the intro), though in general its thought that the decrease in food intake is meant to modulate some perceived stress (whether real, fear-related, or anxiety-based). The abstract similarly makes a claim that in anorexia, "anxiety dominates over hunger" and that is an oversimplified view of the clinical complexity. The intro similarly oversimplifies the findings of the feeding/emotion studies based in the amygdala and lateral hypothalamus, which leads to potential misinterpretation and does not do justice to that body of work, nor does it come across as relevant to the results of this paper (there is no mention of how the PVHLSV circuitry may relate to the known circuitry, i.e. LSVLH or amygdala). The introduction and conclusion would benefit from significant re-writing.

Response: We appreciate the reviewer for raising the concern on how to integrate our findings with literature. As the reviewer also pointed out, many of these points are still a topic of debate. We review that our data presented here reveal an important connection between PVH, a known feeding center, and LS, a known brain regions for emotion, and our functional data convincingly demonstrate that activation of PVH→LSv projections promotes a state of negative valence associated with increased stress/anxiety/fear and at the same time potently inhibits feeding. Within the literature, emerging studies in mice suggest that the neural pathways (AgRP neurons, LH neurons, Parabrachial neurons etc) known to regulate feeding also play a role in the regulation of valence associated with changes in emotional states such as anxiety, aggression and fear; and that neural pathways (amygdala) known to regulate emotion also play a role in the regulation of feeding. Importantly, a large body of human literature suggest a tight

association between feeding abnormalities and emotional defects such as anorexia with heightened anxiety. Notably, treatment of human patients with anti-psychiatric diseases with a goal to reduce anxiety is associated with increased feeding, suggesting a potential interaction at the circuit level in the regulation of feeding and emotional states. Specifically, for both PVH and LS, PVH neurons, especially CRH neurons, are known to be involved in stress responses, and there are some studies in LS, especially involving GLP1R function on a potential role in feeding. No functional connection has been made between PVH and LS.

Under this backdrop, we think it is appropriate for us to integrate our findings on the PVH→LSv projection into the body of work described above on both human and mouse studies. Thus, in Introduction, we first introduced basic facts found in humans on connections between psychiatric disorders and feeding abnormalities, and then introduced current state of research in animals on the same topic to readers. Although we have no data to directly demonstrate that a direct involvement of the PVH→LSv projection in human psychiatric disorders or a direct connection of the projection to those described in other mouse studies cited in Introduction, we think that these are important background pieces of information for readers to know.

Specifically, our data suggest that over-activation of PVH→LSv projection will cause a state of negative valence with exaggerated anxiety/fear-like behaviors, and at the same time, reduce feeding, mimicking the state of anorexia. The general goal of treatment of psychiatric disorders is to reduce anxiety, which, however, is frequently associated with obesity development and increased food intake. Thus, to our knowledge, the underlying biology between the two observations is consistent with each other very well.

Based on the information that provided above, we provide some discussions in Discussion on potential implications of our findings in human disorders with anorexia. We appreciate the view of the reviewer on the debated topic in human literature on causal relationship in abnormalities between feeding and anxiety/stress. However, we have to respectfully point out that it is appropriate for us to use our novel findings with new techniques and more definite animal models to refine and advance the existing model/view, but not to be limited to it, which is exactly the purpose of our ongoing study.

We agree with the reviewer that we need to avoid overstating findings. Based on the reviewer's suggestion, we have modified the wording in Introduction, including deletion of the strong sentence mentioned by the review in Abstract, as well as other changes to avoid overstating the information from citations.

5) Throughout the description of the studies, the authors do not make their methodology clear. For example, for the studies of figure 2, was the optical stimulation unilateral or bilateral? For figure 3, which mice were used? Were both short and long pulses used? For figure 4 it would be useful if the manuscript overtly stated which part of the field was paired to the stimulation. For supplementary figures 6-8 in the results description it would be best to state what was seen in the experiment: the current statement “failed to find evidence for a contribution... through oxytocin, CRH, AVP neurons” is confusing to interpret; it would be more precise and descriptive to say “failed to find anterograde projections” from those neurons. For figure 7, what do different colors mean (different mice? Different days?). In the methods section it says that both male and female mice were used. Given that the PVH is a highly sexually dimorphic region, it would be important to distinguish for what experiments each sex of mouse was used.

Response: We thank the reviewer for pointing out this deficiency and we have revised our manuscript for clarification on all points raised by the reviewer.

6) Finally, I do think the authors may be overstating the novelty of the circuit based investigation. There has been some work on projections from the hypothalamus to the forebrain, though not yet characterized using modern tools (see Sawchenko/Swanson papers from the 80s, and Simerly papers from the 2000s), Furthermore there has been some work done on the reciprocal connections between LSVhypothalamus (Sheehan et al 2004, Sweeney et al 2016). This doesn't dampen the novelty of their work, but I recommend that they temper their description of it to reflect prior work.

Response: We appreciate the reviewer to point out this. Accordingly, we have added the citation of the Swanson paper (9), which describes a general connection of periventricular hypothalamus and lateral septum. However, the Simerly paper ((10), the only one I could find in Pubmed) describes the connection between anteroventral perihypothalamic nucleus (AVPV) with septum, nothing to do with PVH. The Sweeney paper (11) is on the projections of the dorsal part of lateral septum to lateral hypothalamus, not a reciprocal connection, not relevant to these studies either. The Sheehan paper (12) is a review paper discussing functions of lateral septum in behaviors through various efferent projections, not directly relevant to our results. However, these papers have been cited as background information.

Comments from Reviewer 4

In this manuscript, Xu and co-workers investigated a novel neurocircuit underlying the interplay between stress-related and feeding behaviours. They nicely mapped and characterized a novel neurocircuit originating from glutamatergic neurons located in the PVH that project to GABAergic neurons in the LS to control fear and appetite. This manuscript provides interesting and novel data regarding the neurocircuits controlling feeding depending on the emotional responses. This findings will generate a strong interest in the field of research. Overall, the experiments appear well executed (including statistical analysis) and conclusions drawn by the authors are supported by experiments provided in the paper.

Response: We thank the reviewer for the appreciation of our work.

1. As the authors always compared weak (10ms) and strong stimulation (100ms), for consistency, both should be included for all experiments throughout the manuscript. For instance, in Figure 5 the feeding is not depicted for the 100ms stimulation after 6hours fasting. This aspect will also be interesting for the supplementary figures 6 and 7 to rule out that distinct populations could mediate the behaviour observed with the 10ms stimulation vs the 100 ms.

Response: We have added data on 100ms stimulation after 6 hours fasting in Fig. 5. For supplementary figures 6 and 7, 10ms stimulation produced no responses and the results not presented. We have also added a statement "the 10ms stimulation failed to result in any response in mice" in Results.

2. In figure 3d-e, the author should also provide the behavior experiments with optogenetics + CNO only as well as optogenetics + AAV-Fas-ChR2 only.

Response: In Fig 3e-f, the saline group is the optogenetics +AAV-Fax-ChR2 only group. I guess the reviewer is concerned about potential non-specific effect of CNO. In an independent experiment, we confirmed that CNO has no effect on self-grooming/jumping elicited by photostimulation (Supplementary Figs. 4i and 4j). We have added “We confirmed that CNO alone caused no effect on self-grooming or jumping elicited by photostimulation of PVH→LSv fibers” in Result.

3. Figure 5 a-b: would stimulation also change the steady food intake (i.e in random fed animals as in 5f/h) ?

Response: Yes, the photostimulation will also reduce feeding mice. However, since feeding in fed mice is small in the control group with a short time window for photostimulation, the effect on reducing feeding will not be fully appreciated. This is why we used fasting refeeding paradigm in Fig. 5.

4. In general, details are missing in the main text to improve the readability. For instance, when the author describes the figure 3d-f, the mouse line used is not included in the main text but only in the legends of the supplementary figures.

Response: We have carefully read and clarified description in the manuscript including the ones specified by the reviewer.

5. In supplementary figure 1, the annotation “wild-type mice n=14” is confusing/misleading.

Response: We have clarified this point in the Supplementary Fig 1 figure legend.

6. In supplementary figure 3, a “light-OFF” control should be included to compare the c-Fos induction to basal. This aspect is important to further understand the differences observed with 10 and 100ms stimulation.

Response: We have added this data point to the figure.

7. The c-Fos immunostaining depicted in Figure S3 and S4 cannot be visualized on the representative pictures.

Response: The image of the immunostaining has been improved.

8. More discussion on the potential PVH neuronal phenotype involved neurons could be included.

Response: We have added discussion on more PVH neuron groups including GLP1-R neurons, previously shown to be able to regulate feeding.

1. H. Lee *et al.*, Scalable control of mounting and attack by Esr1+ neurons in the ventromedial hypothalamus. *Nature* **509**, 627-632 (2014).
2. B. J. Duistermars, B. D. Pfeiffer, E. D. Hoopfer, D. J. Anderson, A Brain Module for Scalable Control of Complex, Multi-motor Threat Displays. *Neuron* **100**, 1474-1490 e1474 (2018).
3. K. M. Tye, Neural Circuit Motifs in Valence Processing. *Neuron* **100**, 436-452 (2018).
4. V. Ferretti, F. Papaleo, Understanding others: Emotion recognition in humans and other animals. *Genes Brain Behav* **18**, e12544 (2019).
5. T. E. Anthony *et al.*, Control of stress-induced persistent anxiety by an extra-amygdala septohypothalamic circuit. *Cell* **156**, 522-536 (2014).
6. S. Y. Kim *et al.*, Diverging neural pathways assemble a behavioural state from separable features in anxiety. *Nature* **496**, 219-223 (2013).
7. P. S. Kunwar *et al.*, Ventromedial hypothalamic neurons control a defensive emotion state. *Elife* **4**, (2015).
8. L. Wang, I. Z. Chen, D. Lin, Collateral pathways from the ventromedial hypothalamus mediate defensive behaviors. *Neuron* **85**, 1344-1358 (2015).
9. P. Y. Risold, L. W. Swanson, Connections of the rat lateral septal complex. *Brain Res Brain Res Rev* **24**, 115-195 (1997).
10. E. K. Polston, R. B. Simerly, Ontogeny of the projections from the anteroventral periventricular nucleus of the hypothalamus in the female rat. *J Comp Neurol* **495**, 122-132 (2006).
11. P. Sweeney, Y. Yang, An Inhibitory Septum to Lateral Hypothalamus Circuit That Suppresses Feeding. *J Neurosci* **36**, 11185-11195 (2016).
12. T. P. Sheehan, R. A. Chambers, D. S. Russell, Regulation of affect by the lateral septum: implications for neuropsychiatry. *Brain Res Brain Res Rev* **46**, 71-117 (2004).

Reviewers' Comments:

Reviewer #1:

None

Reviewer #2:

Remarks to the Author:

The major issues raised during initial review have been addressed to an adequate enough extent that the manuscript is acceptable, with one exception.

Previous work has indicated that most AAV serotypes (including AAV1) undergo retrograde transport: HUMAN GENE THERAPY 18:195–206 (March 2007). Although there is evidence that AAV1 can move anterogradely, it is not possible to distinguish between anterogradely vs. retrogradely labeled neurons in regions that share reciprocal connectivity (as is the case in LSV and PVN).

The authors should therefore remove panels in Fig. 8i-l (AAV1-Cre to PVN and DIO-Cherry to LS) and remove this section from the text as well.

Reviewer #3:

Remarks to the Author:

This is very interesting work in a relatively untapped field, with well controlled studies showing the novel finding that specifically glutamatergic input from PVH—>LSV bidirectionally affects feeding and anxiety like behavior. The authors should be commended for succinctly summarizing their vast amount of work spanning pharmacology, circuitry, and behavior studies, good use of appropriate controls for most of the studies, and the novelty of the finding. The videos are helpful and the behavior change in response to stimulation is striking. The focus on glutamate is particularly interesting, and timely in relation to the psychiatric neuroscience shift of focus from neuropeptide to neurotransmitter influence on disease, though the authors do not seem to focus on this finding. They instead focus on the novelty of the circuit, and what this means for human anorexia, which is misleading, given that their data shows it is more likely multiple small circuits that together account for the behavior they show, and their understanding of anorexia is incomplete (unfortunately literature on anorexia does not do justice to the heterogeneity and complexity of human illness). I appreciate the authors correction of anthropomorphizing terms and overstated human-relevance throughout the manuscript. Their results remain interesting, important, and absolutely relevant to human disease, but are still inconsistent with the conclusions as stated.

Overall, the authors have done a significant amount of work in response to reviewers comments, but their response to the comments and incorporation of their work into the new version of the manuscript is incomplete. Their new tracing studies, which do provide helpful distinction of the regions involved, do not incorporate reviewers concerns about subpopulations in the LSV, and are not at all integrated into their overall story. Their discussion of the new findings localizing the PVH projections to a small region of the PVH (which is potentially consistent with the AVPV as well) should be incorporated with the tracing studies (for example, those related to CRHR1) and may indicate multiple distinct circuits are responsible for the multiple behaviors observed in their experiments using Sim1-cre or intersectional viruses. Yet the authors give very little time to this, or their difficulty accessing subpopulations of LSV neurons in their discussion.

The conclusions about anorexia, feeding behavior, and anxiety in humans still do not make sense given what is known about anxiety disorders AND anorexia in humans. The authors state that this

circuitry, which both drives anxiety-like response AND suppresses feeding, recapitulates an anorexia like behavior. As the authors are aware, anorexia is commonly co-morbid with anxiety, but it is important to note that suppression of feeding in humans with anorexia is often a way to mitigate anxiety, and patients in the setting of food restriction due to anorexia report less anxiety than when they are satiated or eating. Thus the suppression of feeding would be consistent with a decrease in anxiety response. The authors findings show the opposite of this: the strong negative valence induced by PVHLSV stimulation is associated with decreased food intake. As mentioned by another reviewer, the finding that animals eat less when they display fear-response like behavior is not novel, nor does it recapitulate anorexia. More novel is the fact that inhibition of this circuit leads to feeding. That leaves far less room for interpretation than does the finding of feeding suppression. This could be improved upon if they more clearly state their conclusion that stimulation of the PVH LSV circuitry promotes stronger negative valence than negative valence associated with common assays for fear-response/anxiety-like behavior, as they report in their rebuttal, and do not try to over-stretch the connection to anorexia. A more appropriate model may be to look to the other side of the eating spectrum, related to Prader-Willi or binge eating disorder, to relate that inhibition of these LSV-projecting PVH neurons reduces the PVHLSV excitatory circuit they have observed, allowing for food intake (perhaps overcoming it even in the setting of external stressor since the negative valence associated with the circuitry is so strong).

Minor specific points: Moving more in response to stimulation of the circuit, in combination with changes in feeding behavior, might indicate something else entirely than fear/anxiety response. This could indicate a role for the circuit in metabolism, or that the metabolic changes seen may be compensation for this change in locomotor activity, and should not be regarded as a simple read out of increased anxiety-like behavior. Given that this group studies metabolism, they can appreciate this is important to comment upon.

The GCaMP tracing show in fig 7 does not seem to represent well the summary statistics of 7i.

There are a number of typos in the revised additions to the manuscript which should be addressed, and as in the first iteration of the manuscript, the models are inconsistently described in the text (page 6 GFP neurons). I appreciate the author's improvement in overall description of models in the text and the figures.

Reviewer #4:

Remarks to the Author:

In the revised version of their manuscript, the authors included several additional experiments which improve the characterization of the PVH -> LS neurocircuits. The methodology now appears in a clear way and new control experiments have been added. The authors also added data of the architecture of the studied neurocircuits, which is a really important add-on to their work and to specify the exact sub-nuclei involved. The revised version of the manuscript clearly increases the quality of their work, nonetheless some aspects of it remain to be improved to support their conclusion.

1. The main troublesome point is to know whether the described neurocircuits is indeed specific to feeding. Indeed, the authors have added the results for experiments using the 10mS and the 100mS photostimulation paradigm (not for all the experiments as requested, again this should be added; for instance are mice not jumping at all with the 10mS protocol or are they jumping less ?). However, while comparing results from the 10mS and the 100mS stimulation protocol, none of the experiments performed with the 10mS photostimulation protocol was associated with a change in feeding regulation. Changes in feeding were exclusively observed with the 100mS stimulation, i.e. when the

mice are constantly jumping. Based on the video recording and on the behavioral quantification, mice stimulated with the 100mS protocol are jumping every few seconds. Could these mice actually engage in any kind of behavior at all ? The fact that the authors did not see any changes with lowest photostimulation protocol are questioning their conclusions regarding the feeding regulatory effects of this circuit. Could the male mice, for instance, engage in reproductive behavior if a female will be place in the cage during stimulation ? To ensure appropriate conclusion, the authors need to show that the changes in behavior is indeed specific for feeding and not due to a general inability to engage in other behaviors than jumping. In the same line, to strengthened their conclusion on feeding, the steady food intake (during light cycle) should be included. In the rebuttal letter, the authors specified that it is indeed different but did not show the data. Is that also different with the 10mS or 50 mS photostimulation ?

2. Along the same line of the first comment, it would have been useful to add the circulating levels of corticosterone concentrations during the 10mS and 100mS photostimulation protocol.

3. The authors added controls but for readability and statistical reasons, controls and stimulated/treated group should be included in the same graph (e.g., figure 6 e and f should be merge as they did for Figure 4c for instance). Some controls are still missing: for instance the food intake without stimulation after a 6 hours fast.

4. The c-Fos immunostaining depicted in Figure S3 and S4 can still not be visualized on the representative pictures. We clearly see the ChR2 fibers, but the panel related to c-Fos (b and e for Figure 3 and, c and g for Figure 4) seem to illustrate a nuclear counterstaining rather than neurons immunoreactivity for c-Fos.

5. As the link to anorexia nervosa is purely speculative it would be better to keep it for the discussion part only.

Responses to reviewers' comments

We would like to thank all reviewers again for their additional concerns, which are largely about the writing (#3) or misunderstanding (#4). Here we provide details on how to revise the writing and explanation of our data. Accordingly, the corresponding changes have been made in the text and marked in red.

Comments from Reviewer #2:

Previous work has indicated that most AAV serotypes (including AAV1) undergo retrograde transport: **HUMAN GENE THERAPY 18:195–206 (March 2007)**. Although there is evidence that AAV1 can move anterogradely, it is not possible to distinguish between anterogradely vs. retrogradely labeled neurons in regions that share reciprocal connectivity (as is the case in LSv and PVN).

The authors should therefore remove panels in Fig. 8i-l (AAV1-Cre to PVN and DIO-Cherry to LS) and remove this section from the text as well.

Response: The LS neurons do not project to PVH, as clearly shown by our data from AAV-FLEX-mCherry injections to LSv of Vgat-Cre mice (panel A, below). No fibers were found in the PVH proper but instead strong GFP-expressing fibers were found in the adjacent anterior hypothalamic area (AHA, panel B below). This data is also consistent with the previous report (Anthony TE et al., 2014, Cell 156, 522-536), which showed the LS CRHR2 neurons project to AHA, but not PVH.

Our current AAV1-Cre data should be viewed in the context of our existing data on PVH projections to the LSv, which have convincingly identified LSv-receiving neurons, representing a strong addition to the overall conclusion of the manuscript.

Comments from Reviewer #3

This is very interesting work in a relatively untapped field, with well controlled studies showing the novel finding that specifically glutamatergic input from PVH—>LSV bidirectionally affects feeding and anxiety like behavior. The authors should be commended for succinctly summarizing their vast amount of work spanning pharmacology, circuitry, and behavior studies, good use of appropriate controls for most of the studies, and the novelty of the finding. The videos are helpful and the behavior change in response to stimulation is striking. The focus on glutamate is particularly interesting, and timely in relation to the psychiatric neuroscience shift of focus from neuropeptide to neurotransmitter influence on disease, though the authors do not seem to focus on this finding. They instead focus on the novelty of the circuit, and what this means for human anorexia, which is misleading, given that their data shows it is more likely multiple small circuits that together account for the behavior they show, and their understanding of anorexia is incomplete (unfortunately literature on anorexia does not do justice to the heterogeneity and complexity of human illness). I appreciate the authors correction of anthropomorphizing terms and overstated human-relevance throughout the manuscript. Their results remain interesting, important, and absolutely relevant to human disease, but are still inconsistent with the conclusions as stated.

Overall, the authors have done a significant amount of work in response to reviewers comments, but their response to the comments and incorporation of their work into the new version of the manuscript is incomplete. Their new tracing studies, which do provide helpful distinction of the regions involved, do not incorporate reviewers concerns about subpopulations in the LSV, and are not at all integrated into their overall story. Their discussion of the new findings localizing the PVH projections to a small region of the PVH (which is potentially consistent with the AVPV as well) should be incorporated with the tracing studies (for example, those related to CRHR1) and may indicate multiple distinct circuits are responsible for the multiple behaviors observed in their experiments using Sim1-cre or intersectional viruses. Yet the authors give very little time to this, or their difficulty accessing subpopulations of LSV neurons in their discussion.

Response: We appreciate the reviewer's guidance on improving the discussion of the manuscript. As matter of fact, the point on the sole of glutamate in mediating the action has been discussed in the second paragraph of page 17. The point on discussion related to PVH subsets of neurons and CRHR1 has already been discussed in the first paragraph of page 17. We didn't put too much emphasis on CRHR1 neurons because 1) the effects on CRHR1 is very mild compared to the whole Sim1 neurons, suggesting a minor role of this subset of neurons; and 2) CRHR1-Cre targets a significant portion of neurons outside the PVH proper, which may cause potential concerns from non-PVH projection neurons (i.e. Sim1-Cre negative neurons). In the first paragraph of page 17 of the Discussion, we also mentioned that, based on our data, we identified the location of LSV-receiving neurons through anterograde tracing and they are GABAergic. To illustrate the "difficult accessing subpopulation of LSV neurons", we have added "Given the inherent variation associated with tracing studies, the number of the traced LSV neurons may be underestimated. Further studies using an unbiased approach will be required to reveal the complete subset of LSV neurons that receive synaptic inputs from PVH neurons."

The conclusions about anorexia, feeding behavior, and anxiety in humans still do not make sense given what is known about anxiety disorders AND anorexia in humans. The

authors state that this circuitry, which both drives anxiety-like response AND suppresses feeding, recapitulates an anorexia like behavior. As the authors are aware, anorexia is commonly co-morbid with anxiety, but it is important to note that suppression of feeding in humans with anorexia is often a way to mitigate anxiety, and patients in the setting of food restriction due to anorexia report less anxiety than when they are sated or eating. Thus the suppression of feeding would be consistent with a decrease in anxiety response. The authors findings show the opposite of this: the strong negative valence induced by PVHLSV stimulation is associated with decreased food intake. As mentioned by another reviewer, the finding that animals eat less when they display fear-response like behavior is not novel, nor does it recapitulate anorexia. More novel is the fact that inhibition of this circuit leads to feeding. That leaves far less room for interpretation than does the finding of feeding suppression. This could be improved upon if they more clearly state their conclusion that stimulation of the PVH LSV circuitry promotes stronger negative valence than negative valence associated with common assays for fear-response/anxiety-like behavior, as they report in their rebuttal, and do not try to over-stretch the connection to anorexia. A more appropriate model may be to look to the other side of the eating spectrum, related to Prader-Willi or binge eating disorder, to relate that inhibition of these LSV-projecting PVH neurons reduces the PVHLSV excitatory circuit they have observed, allowing for food intake (perhaps overcoming it even in the setting of external stressor since the negative valence associated with the circuitry is so strong).

Response: The core concern from this reviewer expressed here is the issue of direct comparison to human anorexia. In our original text, we only have two places where we mentioned anorexia: one is in the Introduction and the other is in the last paragraph of the Discussion. To address this, we have eliminated all mentioning of “anorexia” in the text and instead replaced it with a general term of eating disorder”. This general term will still reflect the relevance of our findings to human diseases, as suggested by this reviewer, and at the time avoid a direct comparison to anorexia. It will also help to keep to be consistent with our earlier findings (Mangieri LR et al., Nature Communications, 2018), from which this study is directly involved and in which we discussed relevance of the findings to human eating disorders and anorexia.

Minor specific points: Moving more in response to stimulation of the circuit, in combination with changes in feeding behavior, might indicate something else entirely than fear/anxiety response. This could indicate a role for the circuit in metabolism, or that the metabolic changes seen may be compensation for this change in locomotor activity, and should not be regarded as a simple read out of increased anxiety-like behavior. Given that this group studies metabolism, they can appreciate this is important to comment upon.

Response: As this study is not focused on energy expenditure, there are no data that allow us to discuss on the point of locomotion and metabolism. Our optogenetic studies only involves acute and short periods of time in changes in locomotion, which will not produce any meaningful changes in metabolism.

The GCaMP tracing show in fig 7 does not seem to represent well the summary statistics of 7i.

Response: We have corrected. This is due to a distortion in the y axis of the representative trace during preparation of the figure panel.

There are a number of typos in the revised additions to the manuscript which should be addressed, and as in the first iteration of the manuscript, the models are inconsistently described in the text (page 6 GFP neurons). I appreciate the author's improvement in overall description of models in the text and the figures.

Reponses: We thank the reviewer for carefully reading our manuscript. All of the typos have been corrected.

Comments from Reviewer #4

In the revised version of their manuscript, the authors included several additional experiments which improve the characterization of the PVH -> LS neurocircuits. The methodology now appears in a clear way and new control experiments have been added. The authors also added data of the architecture of the studied neurocircuits, which is a really important add-on to their work and to specify the exact sub-nuclei involved. The revised version of the manuscript clearly increases the quality of their work, nonetheless some aspects of it remain to be improved to support their conclusion.

1. The main troublesome point is to know whether the described neurocircuits is indeed specific to feeding. Indeed, the authors have added the results for experiments using the 10mS and the 100mS photostimulation paradigm (not for all the experiments as requested, again this should be added; for instance are mice not jumping at all with the 10mS protocol or are they jumping less ?). However, while comparing results from the 10mS and the 100mS stimulation protocol, none of the experiments performed with the 10mS photostimulation protocol was associated with a change in feeding regulation. Changes in feeding were exclusively observed with the 100mS stimulation, i.e. when the mice are constantly jumping. Based on the video recording and on the behavioral quantification, mice stimulated with the 100mS protocol are jumping every few seconds. Could these mice actually engage in any kind of behavior at all ? The fact that the authors did not see any changes with lowest photostimulation protocol are questioning their conclusions regarding the feeding regulatory effects of this circuit. Could the male mice, for instance, engage in reproductive behavior if a female will be place in the cage during stimulation ? To ensure appropriate conclusion, the authors need to show that the changes in behavior is indeed specific for feeding and not due to a general inability to engage in other behaviors than jumping. In the same line, to strengthened their conclusion on feeding, the steady food intake (during light cycle) should be included. In the rebuttal letter, the authors specified that it is indeed different but did not show the data. Is that also different with the 10mS or 50 mS photostimulation ?

Response: The core issue raised here is whether 10ms photostimulation causes effects on feeding. As matter of fact, as clearly shown in our Fig. 5a and Fig. 6h, 10ms stimulation reduced feeding. This important data was somehow missed by this reviewer.

2. Along the same line of the first comment, it would have been useful to add the circulating levels of corticosterone concentrations during the 10mS and 100mS photostimulation protocol.

Response: We don't think this measurement will yield a meaningful addition to this study. First of all, we have ruled out a role of PVH CRH neurons in the observed behaviors in our study (Supplementary Fig. 7), suggesting that the behaviors we observed do not directly involved the HPA axis. Second, knowing the corticosterone (cort) level will not add any supportive information to our current conclusion. If the cort level doesn't change, it will suggest that PVH to LSv circuit drives behavior independent of HPA axis (as expected); and if there is a difference, it may suggest that the cort change may be due to a secondary effect from the behavior. So either results will not prove or disapprove our findings. Third, our behavioral changes occur within a few seconds after photostimulation, it is impossible for a direct contribution from potential changes in corts to the observed behaviors. Finally, this question is secondary to the concern raised by this reviewer on the effect of 5Hz 10ms photostimulation on feeding. Since this concern has no basis, there is no necessary to address associated concerns (i.e. cort levels).

3. The authors added controls but for readability and statistical reasons, controls and stimulated/treated group should be included in the same graph (e.g., figure 6 e and f should be merge as they did for Figure 4c for instance). Some controls are still missing: for instance the food intake without stimulation after a 6 hours fast.

Response: We have added conditions of no light in the food intake study with 6 hrs fasting, which was inadvertently missed during preparation of the figure.

4. The c-Fos immunostaining depicted in Figure S3 and S4 can still not be visualized on the representative pictures. We clearly see the ChR2 fibers, but the panel related to c-Fos (b and e for Figure 3 and, c and g for Figure 4) seem to illustrate a nuclear counterstaining rather than neurons immunoreactivity for c-Fos.

Response: We want to clarify that the blue nuclear staining is for c-Fos and we didn't perform nuclear counterstaining in this study. If this is nuclear counterstaining, the blue signal should be universal for the all area in the section; however, as shown in Fig. S4, the area with strong expression of DREADD exhibited much reduced signal, suggesting a specific response to the CNO-DREADD action.

5. As the link to anorexia nervosa is purely speculative it would be better to keep it for the discussion part only.

Response: As suggested Reviewer 3, the mentioning of anorexia has been eliminated.

Reviewers' Comments:

Reviewer #2:

Remarks to the Author:

The concern raised in the last round of review was as follows:

"Previous work has indicated that most AAV serotypes (including AAV1) undergo retrograde transport: HUMAN GENE THERAPY 18:195–206 (March 2007). Although there is evidence that AAV1 can move anterogradely, it is not possible to distinguish between anterogradely vs. retrogradely labeled neurons in regions that share reciprocal connectivity (as is the case in LSv and PVN).

The authors should therefore remove panels in Fig. 8i-l (AAV1-Cre to PVN and DIOCherry to LS) and remove this section from the text as well."

The authors' response to this concern indicates that they do not understand the caveats and limitations of viral tracing. In order for their data to be even potentially interpretable, the AAV1 injection would have to be completely restricted to the PVN. The reason is that if any AAV1 gets outside of the PVN, it would be impossible to know whether any observed anterograde or retrograde labeling occurred due to infection of neurons within the PVN or surrounding regions. In fact, Figure 8j shows the pattern of AAV1 Cre expression, and it is clear that AAV1 delivery was not restricted to the PVN, as Cre+ neurons are present throughout periventricular regions outside the PVN. Therefore, the neurons labeled in the LS shown in Figure 8k and 8l could be: 1) LS neurons that project to PVN, retrogradely labeled via this projection; 2) LS neurons that project to other hypothalamic regions, retrogradely labeled by AAV1 that spread to these extra-PVN areas; 3) LS neurons that are postsynaptic targets of these extra-PVN regions, labeled by anterograde movement of the virus; or 4) LS neurons that are postsynaptic targets of PVN neurons.

It would be a disservice to the field if such data were published, as other investigators unfamiliar with these issues might read the paper and attempt similar flawed experiments.

Reviewer #3:

Remarks to the Author:

I appreciate the authors revisions in response to reviewer comments and recommendations; this revision has improved the quality of the manuscript, and I look forward to further scientific discussion.

Reviewer #4:

Remarks to the Author:

The authors replied to the comments raised previously. I only have two minor points that could be improved.

* Accordingly to my previous comment #1 related to the feeding effects using the 10mS, the authors indeed showed changes in feeding after the 10mS protocol. I would nonetheless strongly suggest discussing whether the PVH->LS pathway is a specific feeding regulatory neurocircuit or a neurocircuit that potentially control "stress-emotional responses" and therefore suppress any other behaviour, in this case feeding. The fact that inhibiting the PVH-> LS projection increases feeding indeed suggest that it could be a feeding specific feeding-regulatory pathway but it will be important to discuss this point in

discussion.

* Regarding the cFos, while the quantification appears very convincing, the representative pictures much less. The number of cFos positive cells seems very high and it is therefore difficult to see the difference between the 10 and 100 mS, the author should include a representative picture of cFos with light-OFF in figure 3.

Responses to reviewers' comments

Reviewer 2 comment:

“Previous work has indicated that most AAV serotypes (including AAV1) undergo retrograde transport: HUMAN GENE THERAPY 18:195–206 (March 2007). Although there is evidence that AAV1 can move anterogradely, it is not possible to distinguish between anterogradely vs. retrogradely labeled neurons in regions that share reciprocal connectivity (as is the case in LSv and PVN). The authors should therefore remove panels in Fig. 8i-l (AAV1-Cre to PVN and DIOCherry to LS) and remove this section from the text as well.”

The authors' response to this concern indicates that they do not understand the caveats and limitations of viral tracing. In order for their data to be even potentially interpretable, the AAV1 injection would have to be completely restricted to the PVN. The reason is that if any AAV1 gets outside of the PVN, it would be impossible to know whether any observed anterograde or retrograde labeling occurred due to infection of neurons within the PVN or surrounding regions. In fact, Figure 8j shows the pattern of AAV1 Cre expression, and it is clear that AAV1 delivery was not restricted to the PVN, as Cre+ neurons are present throughout periventricular regions outside the PVN. Therefore, the neurons labeled in the LS shown in Figure 8k and 8l could be: 1) LS neurons that project to PVN, retrogradely labeled via this projection; 2) LS neurons that project to other hypothalamic regions, retrogradely labeled by AAV1 that spread to these extra-PVN areas; 3) LS neurons that are postsynaptic targets of these extra-PVN regions, labeled by anterograde movement of the virus; or 4) LS neurons that are postsynaptic targets of PVN neurons.

It would be a disservice to the field if such data were published, as other investigators unfamiliar with these issues might read the paper and attempt similar flawed experiments.

Response: this reviewer cites the paper Taymans J et al., Human Gene Therapy 2007, which describes injection of AAV1-GFP to striatum and some positive neurons were also observed in globus pallidus, but not in substantia nigra (only one neuron shown in panel P) and uses this as evidence that AAV1 can move in a retrograde fashion. However, after a careful examination of the data, we found some inconsistency: 1) globus pallidus is not a major site that sends projection to striatum, but showed a lot of neurons with “retrograde” traced neurons; 2) substantia nigra is the well-established site sending abundant projection to the striatal neurons with collaterals to almost all neurons in the striatum, yet only one neuron was labelled; 3) the authors described the volume of injection is 2ul (, which is too big to achieve specific targeting of striatum, and will likely to spill over to nearby globus pallidus; 4) later other studies using AAV1 didn't report this phenomenon of massive retrograde tracing of AAV1 (Aschauer DF et al., Plos one, 2013, V8 e76310; McFarland NR et al., J. Neurochem, 2009 109(3), 838-845). These collective data indicate the evidence of AAV1 retrograde tracing is weak and unfounded.

In contrast, the anterograde tracing of AAV1 vector was described recently by Zingg B et al, Neuron, 2017, 93: 33-47 is strong and verified with careful electrophysiology and behaviors. The unique feature of AAV1-Cre anterograde tracing is that the amount of traced virus is very low due to transsynaptic movement, which will not label downstream

neurons clearly; however, the traced Cre exhibits sufficient Cre activity to mediate reporter expression, as demonstrated in Fig. 2 of the paper. This feature of anterograde tracing is distinct from the retrograde tracing, which doesn't require transsynaptic movement, and will therefore label individual upstream neurons clearly.

Specifically for our case, we injected AAV-Cre-GFP to PVH (Fig. 8j), but didn't observe any clearly GFP-labelled neurons in lateral septum (Figs. 8k and 8i); however, injection of Cre-reporter viruses (AAV-DIO-mCherry) demonstrated Cre-mediated reporter expression in lateral septum (Figs. 8k and 8i).

Importantly, we have evidence that lateral septum doesn't project to PVH, as shown clearly in Fig. 1 below. We delivered AAVdj-FLEX-mCherry to lateral septum (LS) of Vgat-Cre mice (Fig. 1A) and found no mCherry fibers in the PVH (Fig. 1B, circled area) but with abundant fibers in anterior hypothalamic area (AHA). The data suggest that LSv neurons do not project to PVH and the LS projection to AHA is consistent with the earlier publication (Anthony T et al., Cell, 156: 522-536, 2014) where functional mapping of LS to AHA is described.

To further verify whether PVH receives projections from LS, we injected retrograde AAVrg-Cre-GFP to the PVH (Fig2, below, bottom panels), and we found clearly labelled neurons in SCN (i.e. retrograde labelled neurons), but essential no neurons labelled in LSv. However, the AAV1-Cre-GFP injection to PVH showed no SCN neurons labelling (Fig2, below, top panels), suggesting no retrograde tracing activity of AAV1 vectors. In contrast, AAV1-Cre-GFP injections to the PVH yields some GFP positive structures in LSv with Cre-activity in some LSv neurons as evidenced from Cre-dependent reporter expression (AAV-DIO-mCherry). This data are consistent with the tracing data demonstrated in Zingg B et al, Neuron, 2017, 93: 33-47 and clearly demonstrate that AAV1 can effectively trace downstream neurons in an anterograde fashion, but not in a retrograde fashion.

Based on these data and with our existing data showing monosynaptic glutamatergic projections from PVH to LSv, I think there is no concern regarding our AAV1 data, which will add significantly to the described identification of a novel PVH projection to LSv.

Comments from Reviewer #4

1) Accordingly to my previous comment #1 related to the feeding effects using the 10mS, the authors indeed showed changes in feeding after the 10mS protocol. I would nonetheless strongly suggest discussing whether the PVH->LS pathway is a specific feeding regulatory neurocircuit or a neurocircuit that potentially control "stress-emotional responses" and therefore suppress any other behaviour, in this case feeding. The fact that inhibiting the PVH-> LS projection increases feeding indeed suggest that it could be a feeding specific feeding-regulatory pathway but it will be important to discuss this point in discussion.

Response: the significance of feeding has already been discussed.

2) Regarding the cFos, while the quantification appears very convincing, the representative pictures much less. The number of cFos positive cells seems very high and it is therefore difficult to see the difference between the 10 and 100 mS, the author should include a representative picture of cFos with light-OFF in figure 3.

Response: We have included a section on c-Fos at baseline no-stimulated conditions.